# Measurement report: The promotion of low-level jet and thermal-effect on development of deep convective boundary layer at the southern edge of the Taklimakan Desert

Lian Su[1], Chunsong Lu[2,3], Jinlong Yuan[2,3], Xiaofei Wang[4], Qing He[5], Haiyun Xia[1,2,3,*]

[1]School of Earth and Space Science, University of Science and Technology of China, Hefei 230026, China
[2]School of Atmospheric Physics, Nanjing University of Information Science and Technology, Nanjing 210044, China
[3]Collaborative Innovation Center on Forecast and Evaluation of Meteorological Disasters, Key Laboratory for Aerosol-Cloud-Precipitation of China Meteorological Administration, Key Laboratory of Meteorological Disaster of Ministry of Education,
NUIST, Nanjing 210044, China
[4]Xinjiang Uygur Autonomous Region Meteorological Service, Urumqi 830002, China
[5]Institute of Desert Meteorology, China Meteorological Administration, Urumqi 830002, China

*Correspondence to*: Haiyun Xia (hsia@ustc.edu.cn)

**Abstract.** A vigorous development process of the deep convective boundary layer (CBL) was observed at the southern edge
of the Taklimakan Desert on 6 June, 2022. Based on coherent Doppler wind lidar and ERA5 data, the formation mechanism of the deep CBL exceeding 5 km was analyzed, which was mainly driven by the low-level jet (LLJ) and thermal-effects. During the stage of LLJ preceding the formation of the deep CBL, the LLJ had adequately prepared the conditions for the development of the deep CBL in terms of momentum, energy, and material. Firstly, the cold downhill airflow from the Tibet Plateau, which leads to the formation of LLJ, weakens the height and intensity of the temperature inversion layer, thereby
reducing the energy demand for the breakdown of this layer. Secondly, the LLJ not only supplements the material and energy in the residual layer, but also suppresses the exchange with the lower atmosphere. In addition, the LLJ provides a driving force for the development of the deep CBL. During the stage of thermal effects, the sensible heat driven air-pump from the Tibet Plateau and the passage of a cold front provide additional impetus for the development of the deep CBL. Finally, the formation of deep CBL was catalyzed by extreme thermal effects of the underlying surface, such as the furnace effect and the atmospheric
superadiabatic expansion process. The study of the development of the deep CBL is important for revealing the land-air exchange process of momentum, energy, and material between the Taklimakan Desert and the Tibetan Plateau.


# 1 Introduction

The atmospheric boundary layer (ABL) serves as the interface where the earth's surface exchanges momentum, energy and material with the free atmosphere (Stull, 1988; Garratt, 1994). The boundary layer height (BLH) is an important meteorological reference variable in the vertical direction, which indicates the atmospheric environmental capacity of the region and the vertical diffusion degree of pollutants (Holtslag and Boville, 1993). Studying in the temporal and spatial distribution of BLH, which is closely linked to human life, plays a crucial auxiliary role in monitoring air pollution and formulating pollution control policies tailored to local conditions.

The convective boundary layer (CBL) belongs to an unstable ABL, and the height of CBL should usually be lower than 2-3 km. However, under specific conditions such as arid regions and monsoon climates, the height of CBL can continue to develop upwards and may exceed 5 km (Garratt, 1994). At present, a large number of scholars have found and analyzed the deep CBL phenomenon in the subcontinent of India (Basha and Ratnam, 2009; Raman et al., 1990), Sahara desert (Birch et al., 2012; Marsham et al., 2008), Mongolia (Han et al., 2015), Tibet Plateau (Che and Zhao, 2021; Lai et al., 2023), Badain Jaran Desert (Han et al., 2012), and Gobi desert (Zhang et al., 2002). These studies also revealed that the deep CBL exerts an influence on the local pollutant transmission and diffusion, cloud formation processes, strong convective weather, rainfall, drought and so on. The Taklimakan Desert (TD), which holds a significant role in global climate change, has also carried out corresponding research work. For example, in the hinterland of the TD, the intense surface heating is not the primary factor driving the formation of deep CBL. Instead, the presence of weak temperature inversion and a near-neutral residual layer (RL) above the CBL are crucial factors (Zhang et al., 2022; Xu et al., 2018); The low-level jet (LLJ) can trigger significant air accumulation and dynamic convergence in the lower atmosphere, while the deep CBL is usually accompanied by the LLJ on the following night (Wang et al., 2019); The deep CBL facilitates cloud formation in the late afternoon. This cloud formation not only leads to substantial surface cooling but also causes the momentum in the upper part of the boundary layer to transport downward, resulting in dust emissions (Zhang et al., 2024). The MinFeng station, situated on the northern slope of the Tibet Plateau (TP) and the southern edge of the TD, is a location known for its severe wind-sand activities (Yang et al., 2016; Xiao et al., 2008), and was established in 2018 (Yang et al., 2020). The unique geographical location of the study site (TD, slope terrain, Kunlun Mountains, TP) makes the formation mechanism of the deep CBL not only complex but also highly significant. For example, within the study area, special meteorological phenomena such as drought, severe convective weather, dust storms, gales, low-level jets, wind shear, and others frequently occur concomitantly with the development of the deep CBL (Su et al., 2024b; Wang et al., 2016; Ge et al., 2016). The annual average number of days with dust weather is 113.5 (Yang et al., 2016), and during summer, the number of days with a BLH exceeding 4 km surpasses that observed at other major weather stations within the TD (Wang et al., 2019). Investigating the deep CBL is instrumental in comprehending the formation and evolution of dust pollution weather and contributes to the management of the ecological environment. Furthermore, under the combined influence of the deep CBL and the driving force emanating from the northern slope of the TP, dust aerosols within the study site have the capability to ascend to heights exceeding 7 km (Meng et al., 2019), ultimately impacting regional and potentially

even global precipitation patterns, cloud cover, and material circulation during their long-distance transportation (Ge et al., 2014; Huang et al., 2014).

When conducting experiments in the desert, the harsh desert climate environment exacerbates the performance requirements and the maintenance costs of meteorological equipment, ultimately increasing the difficulty of environmental monitoring. The coherent Doppler wind lidar (CDWL) needs to operate under the conditions of strict sealing and precise temperature control. In the detection of the BLH, the CDWL exhibits characteristics such as low blind area, high radial spatial resolution and temporal resolution, long detection distance and little influence by ground clutter. These characteristics of CDWL enable it to obtain the airflow conditions in the atmosphere from the calculated wind field information, monitor changes in BLH in real-time with greater accuracy, and aid in understanding the diffusion and retention of dust pollutants. Overall, the CDWL is suitable for long-term continuous and stable detection in desert areas, and it is one of the effective methods for estimating the BLH in such environments (Li et al., 2017; Zhang et al., 2020; Collis, 1966; Zhang et al., 2021).

In this paper, the CDWL was utilized to conduct a long-term stable observation experiment in the MinFeng area of the TD. On 6 June, 2022, local time (UTC+8), a representative formation process of the deep CBL was observed. Both CDWL data and ERA5 data were used to analyze the causes of the formation of the deep CBL. This paper is organized as follows: the study site, datasets and methods are described in Sect. 2. The CDWL observation results are presented and analyzed in Sect. 3. From the perspective of the whole desert region, the ERA5 reanalysis data were also analyzed in Sect. 4. Finally, a conclusion is drawn in Sect. 5.

## 2 Site, data resources, methods

### 2.1 Study site

The Taklimakan Desert is the second largest shifting desert in the world and the largest desert of China. Due to the blocking effect of the Tibet Plateau on the warm and humid airflow, the TD has become a typical extreme arid climate zone. The study site of MinFeng (37.06° N, 82.69° E, elevation 1418 m), as shown in Fig.1, located on the southern edge of TD and adjacent to the northern foot of Kunlun Mountains, which is significantly influenced by the TP. The area is characterized by long sunshine hours, intense radiation, scarce precipitation, and the convergence of east-west airflows, making it the site with most frequent wind-sand disasters in China. The average number of dusty days in this area exceeds 113.5 days per year, and the frequent wind-sand weather has a significant impact on human activities and health (Zhou et al., 2020; Zhou et al., 2022; Yang et al., 2016; Wu et al., 2016).

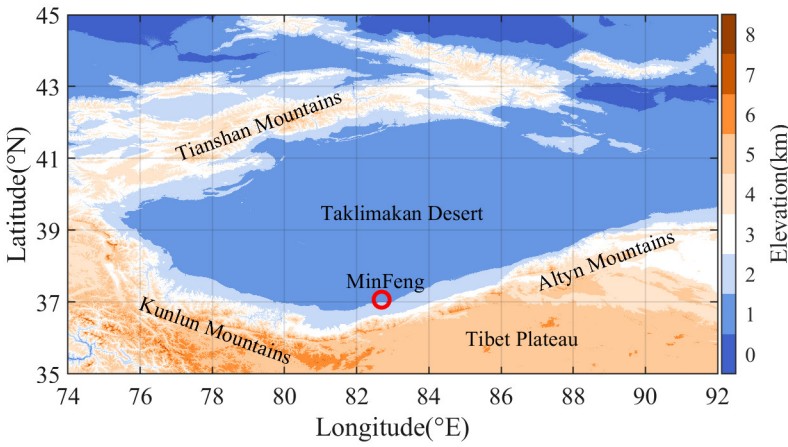

**Figure 1.** Elevation map of the Taklimakan Desert. The red circle represents the study site of MinFeng.

## 2.2 Instruments and dataset

In this study, a compact CDWL working at eye-safe wavelength of 1.5 μm is used. The laser has a pulse energy of 30 μJ and a repetition frequency of 10 kHz. The temporal resolution and radial spatial resolution are 1 minute and 30 m, respectively. During the experiment, the lidar operates in the velocity azimuth display (VAD) scanning mode with an elevation angle of 70°. The key parameters of the CDWL are listed in Table 1.

**Table 1.** Key Parameters of the CDWL

| Parameter | Value |
|---|---|
| Wavelength | 1.5 μm |
| Pulse energy | 130 μJ |
| Pulse repletion frequency | 10 kHz |
| Diameter of telescope | 100 mm |
| Radial spatial resolution | 30 m |
| Azimuth scanning range | 0-360° |
| Zenith angle | 70° |
| Sample rate of ADC | 500 MS/s |

Local meteorological data are provided by the MinFeng County Meteorological Bureau, including air temperature, ground temperature, relative humidity, horizontal visibility, etc.

ERA5 (Hersbach et al., 2020) is the fifth-generation global climate reanalysis dataset from the European Centre for Medium-Range Weather Forecasts (ECMWF). In this paper, ERA5 reanalysis data, such as μ-ν-ω wind vector, atmospheric temperature, relative humidity, mean sea level pressure, surface latent heat flux, boundary layer height, etc. are applied to analyze the

regional variation of the whole desert. The spatial resolution of the reanalysis data is 0.25°×0.25°, and the temporal resolution is 1 hour.

## 2.3 Methods

The data obtained from the CDWL exhibit high radial spatial and temporal resolution, making it suitable for estimating the turbulent kinetic energy dissipation rate (TKEDR) at different heights. The backscatter signal and turbulence intensity detected by CDWL are sharply reduced due to the temperature inversion characteristic of the atmospheric boundary layer top (Hooper and Eloranta, 1986). Based on this, the TKEDR threshold method can effectively estimate the BLH (Wang et al., 2021; Banakh et al., 2021).

The calculation formula of TKEDR is as follows (Banakh and Smalikho, 2018):

$$TKEDR = [\frac{\bar{D}_L(\varphi_l) - \bar{D}_L(\varphi_1)}{A(l\Delta y_k) - A(\Delta y_k)}]^{\frac{3}{2}} \tag{1}$$

where $\bar{D}_L(\varphi_l)$ is azimuth structure function. $L$ is the serial number for the laser beam's line of sight. $\varphi_l = l\Delta\theta$, $\Delta\theta$ is the azimuth angle resolution, and $l=1,2,3\ldots$. The $A(l\Delta y_k)$ is calculated theoretically for the Kolmogorov model of the two-dimensional turbulence spectrum (Banakh et al., 2017), $\Delta y_k$ is the transverse dimension of the probed volume, and $k$ is the range gate number, $k=1,2,3\ldots$. The error analysis for calculating TKEDR and BLH was conducted by Viktor A. Banakh (Banakh et al., 2017; Banakh et al., 2021).

In this experiment, the value of $l$ is set to 2, and the threshold of TKEDR is set to $10^{-4}$ m$^2$ s$^{-3}$. When the location is at the height of $H_n = \Delta R * N$ ($N$ is the index number of bins, and $\Delta R$ is radial spatial resolution), if all TKEDR values within the range $[\Delta R*(N+1), \Delta R*(N+5)]$ are less than the threshold, then $H_n$ is used as the BLH.

The same type of CDWL also realizes the calculation and verification of TKEDR and BLH in various application scenarios (Wang et al., 2021; Wang et al., 2022; Jiang et al., 2022; Yuan et al., 2020; Yuan et al., 2021; Wu et al., 2023; Li et al., 2023).

## 3 Lidar results and local analysis

The underlying surface of the desert causes a significant drop in night temperature, making the formation of the inversion layer (IL) more likely during nighttime. Compared with the nighttime, changes in the mixed boundary layer height during the daytime offer a better reflection of the development of local dust pollution. In this paper, the statistics have been collected on the probability of monthly occurrence of boundary layer at different heights during the daytime (8:00 LT~21:00 LT) from September 2021 to August 2022, with a total of 50663 samples. It can be clearly seen from Figure 2 that a small number of boundary layer heights exceeded 5 km in June 2022. In order to explain this phenomenon, the typical boundary layer data on 6 June, 2022 were selected for analysis.

Fig. 3 displays the continuous observation results obtained from the CDWL, local meteorological equipment, and ERA5 on 6 June, 2022, local time (LT, UTC+8). The CDWL has a radial spatial resolution of 30 m. By analyzing the original data of

power spectrum, various parameters such as CNR, TKEDR, vertical wind speed, horizontal wind speed and horizontal wind direction were derived. The CNR can be used as an indicator of aerosol concentration (Pea et al., 2013). In the preprocessing stage, calculated products are eliminated if the CNR value is below -17 dB. In this paper, the time period from 0:00 LT to 12:00 LT is categorized as the LLJ stage, while 12:00 LT~24:00 LT is classified as the thermal effect stage.

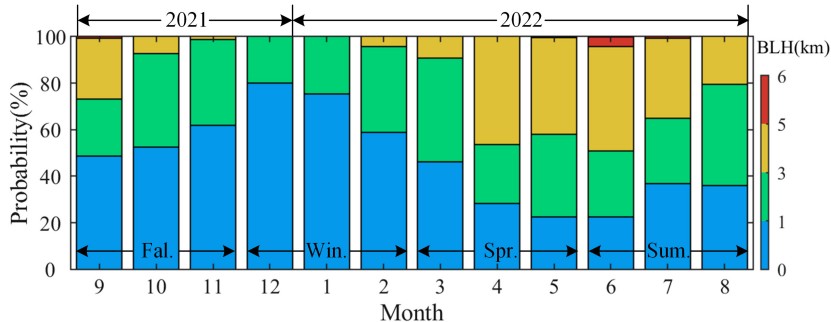

**Figure 2.** The monthly probability distribution of the boundary layer height of MinFeng in the Taklimakan Desert during the daytime from September 2021 to August 2022.

The LLJ generally refers to the strong and narrow airflow zone with wind speed exceeding 12 m s$^{-1}$ within a height of 3 km (Bonner, 1968). As evident from Fig. 3b, there is a clear occurrence of the LLJ phenomenon in the study area before 12:00 LT, with the central axis of the LLJ positioned at approximately 1 km.

At 0:00 LT~6:00 LT, before the formation of the LLJ. It can be clearly found in Fig. 3b that the horizontal wind speed has subsided from 4 km to 2 km, and the wind is the downhill wind blowing from the TP to the desert (Fig .3e, >3 km, 225°-295°). When the downhill airflow was superimposed on the desert background wind field, the horizontal wind speed increased to a maximum of 16.62 m s$^{-1}$ at 5:41 LT. This wind speed exceeded the critical sand-raising wind speed range of 3.5~10.9 m s$^{-1}$ (Yang et al., 2017) and promoted the formation of LLJ (Matsumoto and Ninomiya, 1971; Mcnider and Pielke, 1981). The enhanced turbulence activity near the surface was also observed, and the BLH was stabilized at about 1 km (Fig. 3c). Additionally, according to the local meteorological data, the study site experienced dust weather during this period (Fig. A2).

At 6:00 LT~12:00 LT, this period represents the maintenance stage of the LLJ. As shown in Fig. 3(f-g), during the period of 6:00 LT~10:00 LT, with the cold downhill airflow and the upstream cold airflow traveling to the desert basin where the study site is located, the near-surface temperature dropped sharply and the relative humidity increased significantly. Additionally, the surface weather station also recorded that the difference of the ground-air temperature reached a minimum of -0.8 °C at 8:00 LT. At 6:00 LT~8:00 LT, two notable phenomena occurred. Firstly, as depicted in Fig. 3a, the downward transfer of momentum potentially lifted dust aerosols into the residual layer at approximately 3 km (Washington et al., 2006; Fiedler et al., 2013; Ge et al., 2016), thereby facilitating the replenishment of dust aerosols in the residual layer from the desert hinterland. Secondly, in meteorology, the height of the inversion layer often corresponds to the height of the ABL. Consequently, as illustrated in Fig. 3c, the downhill cold airflow weakened the intensity of the IL, reduced its height, and formed a near-surface IL (also analyzed in Fig. 5(p-t)). Furthermore, at 7:58 LT, the maximum wind speed of 15.04 m s$^{-1}$ was reached. The strong wind shear effect beneath the LLJ serves as a momentum source for turbulent activity, causing intermittent pulsation of

turbulence (Ohya et al., 2006; Mathieu et al., 2005).This, in turn, the BLH can be raised to more than 2 km, potentially enabling partial replenishment of material in the residual layer.

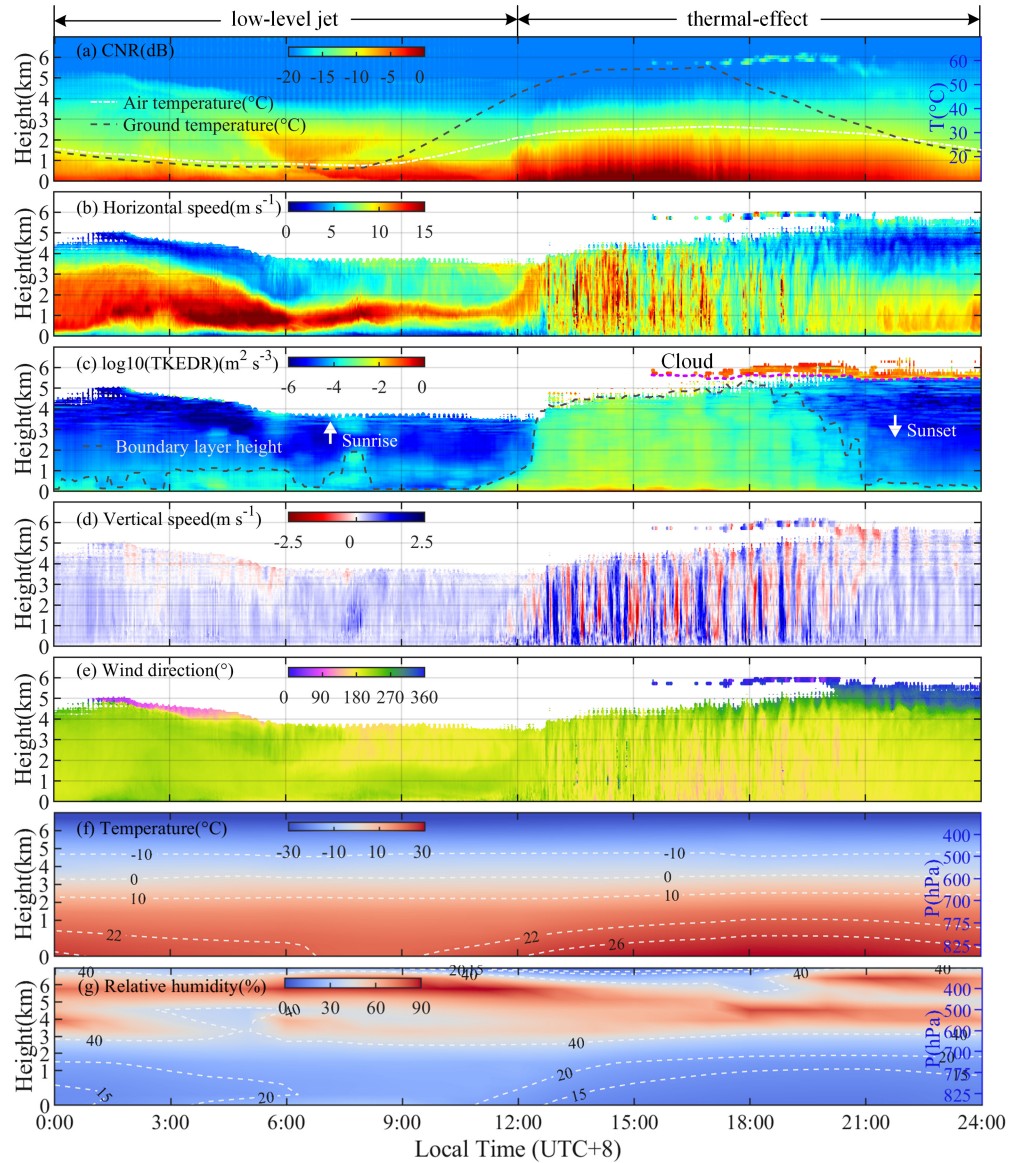

**Figure 3.** The continuous observation results of the CDWL, the local meteorological equipment, and the ERA5 on 6 June, 2022, local time (UTC+8). (a) CNR, the local air and ground temperatures are also shown in this subgraph. (b) horizontal wind speed. (c) log10(TKEDR) and boundary layer height, the time of sunrise and sunset are marked with arrow symbols, and the cloud base height is marked with a purple dotted line. (d) vertical wind speed, the positive vertical wind speed represents the descending speed, and vice versa. (e) horizontal wind direction, 0° represents the wind blows to the north. (f) atmospheric temperature. (g) relative humidity. The height represents the height above the ground of the lidar site. The temperature and relative humidity contours in Fig. 3f and Fig. 3g are denoted by white dashed lines.

During the period from 8:00 LT to 11:00 LT, the LLJ and IL can provide ample momentum, energy and material preparation for the development of the deep CBL. Firstly, as evident from Fig. 3(a-b), due to the existence of the LLJ and the IL, the

atmosphere exhibited a stratified state (Blackadar, 1957). This resulted in a high concentration of CNR values being distributed below 1 km, which serves as a material foundation for boundary layer development. Secondly, the LLJ and the IL play a crucial role in maintaining the balance of atmospheric thermal structure. They inhibit internal turbulent motion and energy exchange with the upper atmosphere, thereby storing enough momentum and energy for the development of deep CBL. Lastly, the weakening of the height and stability of the IL enables the developing boundary layer to reduce the energy of the broken IL, which is conducive to the vigorous development of the subsequent boundary layer.

At 12:00 LT~19:00 LT, with the shortwave radiation of the sun gradually heating the atmosphere and the driving force provided by the LLJ, the stable stratification at low altitude was broken, and the turbulent mixing process began to be reactivated. In Fig.3 (c-d), it evident that the values of TKEDR were consistently maintained at a high level, indicating a significant enhancement in the vertical transport capacity of the atmosphere (Wang et al., 2020). Consequently, the underlying CNR value below 1 km in Fig. 3a increased significantly, and the height of the CBL in Fig. 3c underwent rapid development and exceeded 5 km (Meng et al., 2019). Due to the existence of the LLJ and the IL in the early stage, the dust aerosol was concentrated below 1 km. This lower dust aerosol layer can quickly absorb solar radiation and heat the atmosphere, which can form a "furnace effect" (Ma et al., 2021; Ma et al., 2020) to make the material conditions of the desert boundary layer develop rapidly into thermal conditions. As shown in Fig. 3a and Fig. 3c, the temperature difference of ground-air reached a maximum of 25 °C at 15:00 LT, and the height of the CBL experienced a rapid increase around 13:00 LT. This strong thermal effect significantly promoted the growth of the CBL. Additionally, the LLJ with large wind speed can also provide a basic dynamic condition for the development of the CBL, and the lifting process of the horizontal wind can be obviously observed at 12:00 LT in Fig. 3b. Furthermore, the TP, characterized by its high altitude, thin air, and minimal weakening effect on solar radiation, enables the sensible heat heating of its surface to suck up the surrounding atmosphere, forming a "sensible heat driven air-pump" of the TP (TP-SHAP) (Wu et al., 2012; Wu et al., 2007). The study site of this paper is located at the northern foot of the TP, where the TP-SHAP exerts a significant influence on lifting dust aerosols (Tan et al., 2021). Specifically, the TP-SHAP has the capability to lift dust aerosols from the bottom of the desert along the northern slope of the Kunlun Mountains to the TP (Ge et al., 2014), thus further promoting the development of the local boundary layer (analyzed in Fig. 7). Additionally, the role of continuous hot weather and the entrainment process of the residual layer cannot be ignored (Zhang et al., 2011; Marsham et al., 2008). According to local meteorological data, the hot weather, with temperatures exceeding 30 °C, has persisted for 6 consecutive days.

At 17:00 LT~24:00 LT, the study site gradually became covered by clouds (Fig. 3c, Fig. A1). Before 18:00 LT, the study site was partially obscured by scattered clouds, and after 18:00 LT, the entire study site was fully covered by clouds. The presence of these clouds significantly reduced the solar radiation reaching the surface, causing the surface temperature to decrease rapidly (Fig. 3a), and gradually weakening the turbulence intensity of the atmosphere (Fig. 3c). At 17:00 LT~20:00 LT, the atmospheric turbulence remained active during the initial stages of cloud formation. This was due to two factors: the surface temperature was still significantly higher than the atmospheric temperature, and the heat preservation effect of dust on the atmosphere continued to provide energy for the upper atmosphere. Additionally, the cold clouds moved towards the warm air

mass over the desert, promoting the formation of an upper-level cold front and causing strong convective motion in the lower atmosphere (Fig. 3d). As a result, the height of deep CBL reached its peak at 18:00 LT (Fig. 3c). When the cloud fully covered the study site, the surface radiation further cooled the near-surface air, greatly weakening the atmospheric turbulence intensity and significantly reducing the CBL height before sunset, and the ground-air temperature difference changed to -0.6 °C at 22:00
LT. After the boundary layer developed into a nocturnal stable boundary layer (SBL), the airflow began to recover into a relatively strong and narrow airflow zone (Hoecker, 1963).

The cloud coverage over the Taklimakan Desert is shown in Fig. A1. The local surface meteorological observation data of the day are shown in Fig. A2. At the experimental site, the representative CNR, horizontal wind speed, TKEDR, BLH, vertical wind speed and wind direction in different seasons are also presented in Fig. A3.

## 4 ERA5 results and regional analysis

### 4.1 Low-level jet

The LLJ is closely related to air pollution, dust storm, heavy rainfall and many other aspects. Studying the LLJ can partially reveal the dust emission and transmission process within the study area. Fig. 4 illustrates the variations in wind vector, geopotential height, atmospheric temperature, and relative humidity of the TD at 750 hPa (about 1.05 km above the ground)
from 2:00 LT to 10:00 LT. As depicted in the wind vector subgraphs of Fig. 4(a-e), the study site is situated within the LLJ region, thereby validating the effectiveness of observing LLJ using CDWL. In Fig. 4(a-c), the upstream and downstream of the horizontal airflow correspond to the divergence region (with lower temperature and relative humidity) and the convergence region (with higher temperature and relative humidity), respectively. The horizontal airflow can cause the divergence of the upstream region to sink and the convergence of the downstream region to rise, which is conducive to the development of the
subsequent LLJ and the transport of dust aerosols (Bonner, 1968; Han et al., 2022). The TD is surrounded by three mountains, forming a unique horseshoe-shaped terrain structure. At the northeast of the study site, the northwest wind with lower wind speed deflects to the west after encountering the blocking of the TP. To maintain the conservation of the potential vorticity, an east wind with higher wind speed is formed. The blocking of the mountains further accelerates the formation of the LLJ (Wexler, 1961) and may also contribute to the transport of dust aerosols (Caton Harrison et al., 2021). Compared to the northern
side of the desert, the southern side exhibits higher temperatures, lower geopotential heights, and lower relative humidity, which are conducive to the formation of more active airflow. Under the background conditions of thermal and potential difference, it is helpful to the formation and enhancement of the desert background wind field, such as gradient wind and thermal wind, and to promotes the formation of LLJ (Stensrud, 1996; Rife et al., 2010). During nighttime in the TD, the wind field of the LLJ rotates clockwise and has a typical inertial oscillation phenomenon (Blackadar, 1957). In summary, the LLJ
forms a water vapor convergence zone ahead of the study site and maintains the temperature within a relatively high range (16°C isotherm), enhancing potential instability (Fig. 4(g-i)). This, in turn, strengthens the convective potential of the

atmosphere and provides the essential energy and water vapor conditions for the subsequent development of the boundary layer.

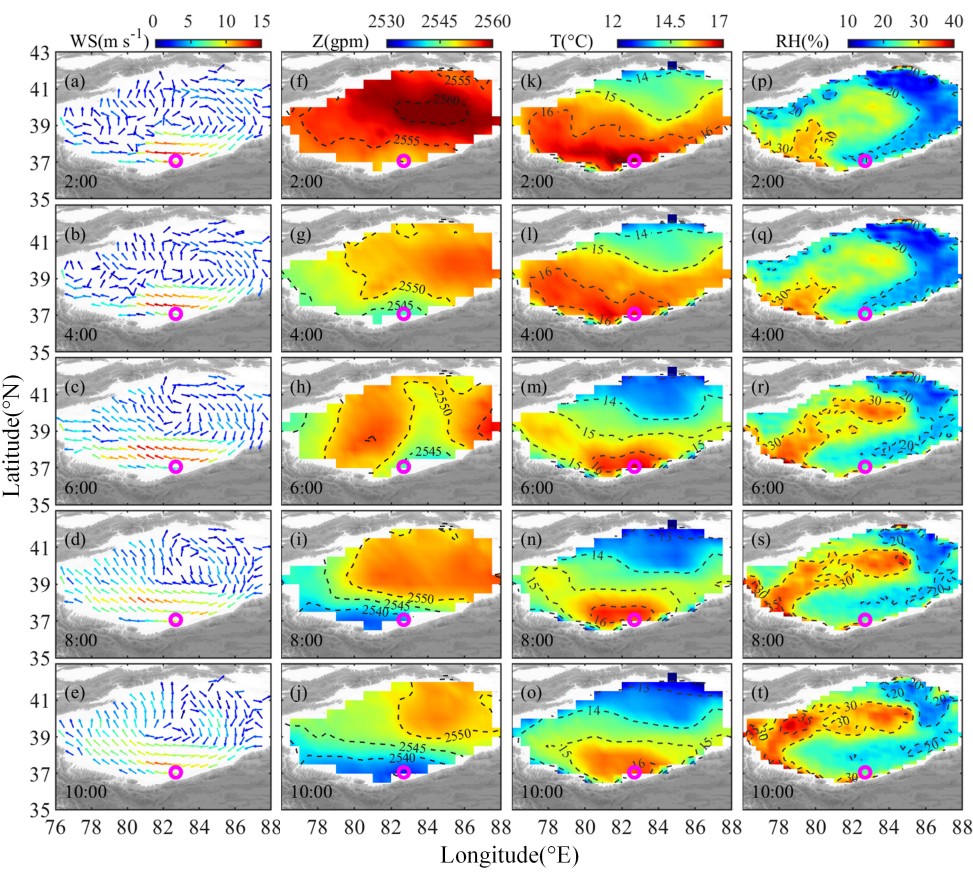

**Figure 4.** The distributions of ERA5 data over the Taklimakan Desert at 750 hPa (about 1.05 km above the ground) from 2:00 LT to 10:00 LT, on 6 June, 2022, local time (UTC+8). (a-e) wind vector. (f-j) geopotential height. (k-o) air temperature. (p-t) relative humidity. The purple circle represents the study site of MinFeng. The contours are denoted by dashed lines.

Fig. 5 illustrates the distribution of mean sea level pressure, surface sensible heat flux, surface latent heat flux, temperature inversion distribution, and BLH. At 2:00 LT~8:00 LT, it can be observed in Fig. 5(a-d) that the pressure over the TP (1025 hPa) is significantly higher than that near the study site in the desert basin (1007 hPa). Consequently, a substantial pressure gradient is established between the TP and the TD, which can facilitate the formation of the downhill airflow under the combined action of gravity and pressure gradient force. When this airflow of the downhill is superimposed on the background wind field of the TD, the wind speed of the TD can be enhanced (similar to Fig. 3b), which also indicates that the influence of topographic baroclinicity is significant (Jones, 2019). During this period, the surface sensible heat flux at the study site is generally exceeds -10 W m$^{-2}$, and the surface latent heat flux is positive. The distribution of these fluxes inhibits convective and turbulent activities within the boundary layer, thereby promoting the development of the nocturnal SBL (Zhang et al., 2017). At the study site, the atmospheric temperature changes induced by the LLJ contribute to the formation of an IL, and the

most obvious IL phenomenon was observed at 8:00 LT. The IL can weaken the convective motion of atmosphere, resulting in the boundary layer height near the study site being constrained within 0.25 km, thereby limiting the diffusion and mixing of dust pollutants, and serving as a source of dust material for further boundary layer development.

At 8:00 LT~10:00 LT, with the increase of solar radiation, the average sea level pressure on the southeast TP gradually decreased, the inversion layer started to dissipate, and the sensible and latent heat fluxes of the desert increased rapidly.

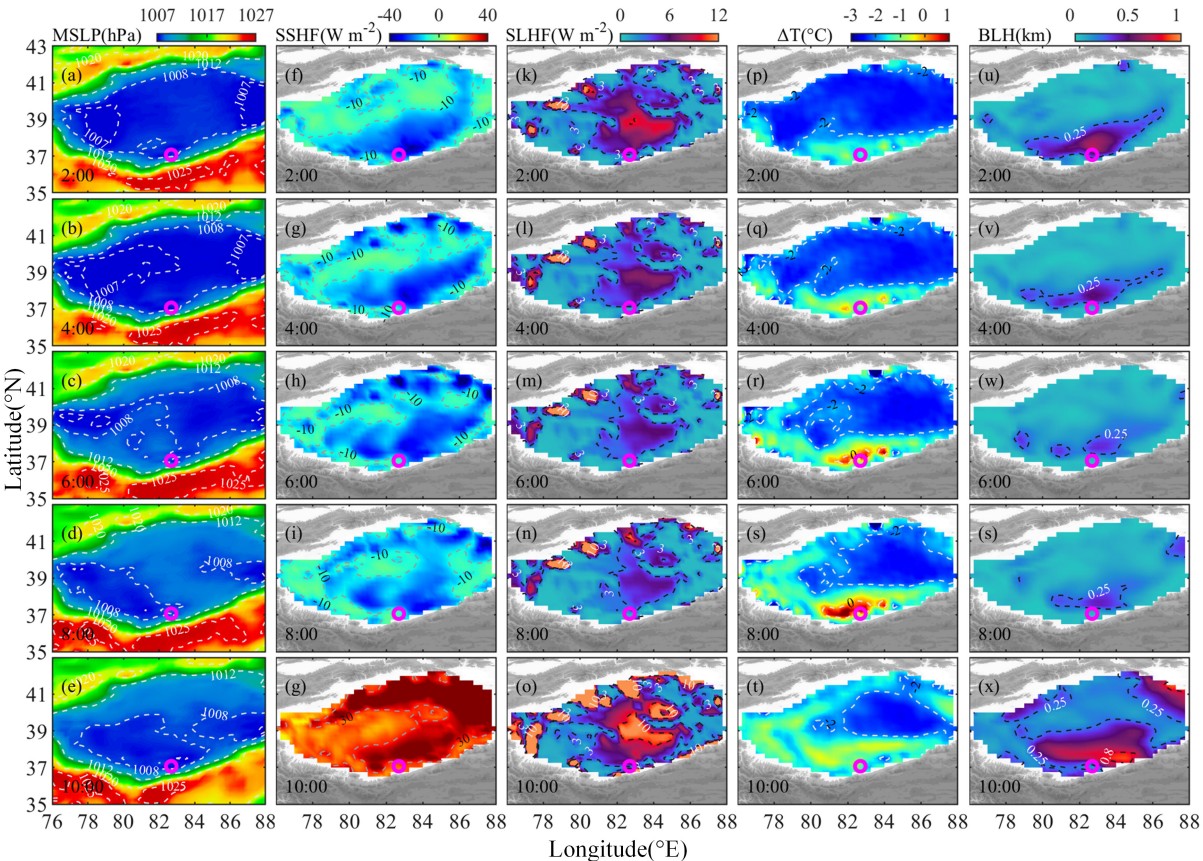

**Figure 5.** The distributions of ERA5 data over the Taklimakan Desert from 2:00 LT to 10:00 LT, on 6 June, 2022, local time (UTC+8). (a-e) mean sea level pressure. (f-j) surface sensible heat flux. (k-o) surface latent heat flux. (p-t) temperature inversion distribution, obtained by subtracting 850 hPa (38 m above the ground) from the temperature data of 825 hPa (232 m above the ground). (u-x) boundary layer height. The purple circle represents the study site of MinFeng. The contours are denoted by dashed lines.

## 4.2 Heat factors

In June, the precipitation in the study region is scarce, the underlying surface is dominated by fine sand, and the soil is dry with strong evaporation capacity. Fig. 6 shows the distribution of wind vector, air temperature, surface sensible heat flux, surface latent heat flux and boundary layer height over the TD from 12:00 LT to 22:00 LT. At 12:00 LT~18:00 LT, with the dissipation of the LLJ, the study site gradually shifted from the east wind to the northeast wind blowing towards the TP. The

persistent high solar radiation resulted in a surface sensible heat flux exceeding 300 W m⁻² near the study site (at 16:00 LT), leading to highly efficient atmospheric heating (Zhang et al., 2002). Firstly, heating the atmosphere with such a high surface sensible heat flux can promote the generation of thermal convection, thereby enhancing atmospheric turbulence and facilitating rapid boundary layer development. Secondly, the atmospheric temperature of the study region consistently occupies the highest value region at each moment (30 °C isotherm), enabling the formation of a s lower low-pressure center that attracts dust from relatively colder areas. Consequently, at 18:00 LT, a deep CBL with a maximum height of 4546 m was formed near the study site. At 20:00 LT~22:00 LT, the surface sensible heat flux swiftly turned negative, and the BLH decreased to less than 1 km. Compared to the CDWL data, the ERA5 reanalysis data exhibit coarser temporal and spatial resolution, and the calculation method for the BLH also differs, resulting in a lower maximum height for the deep CBL when compared to the CDWL data.

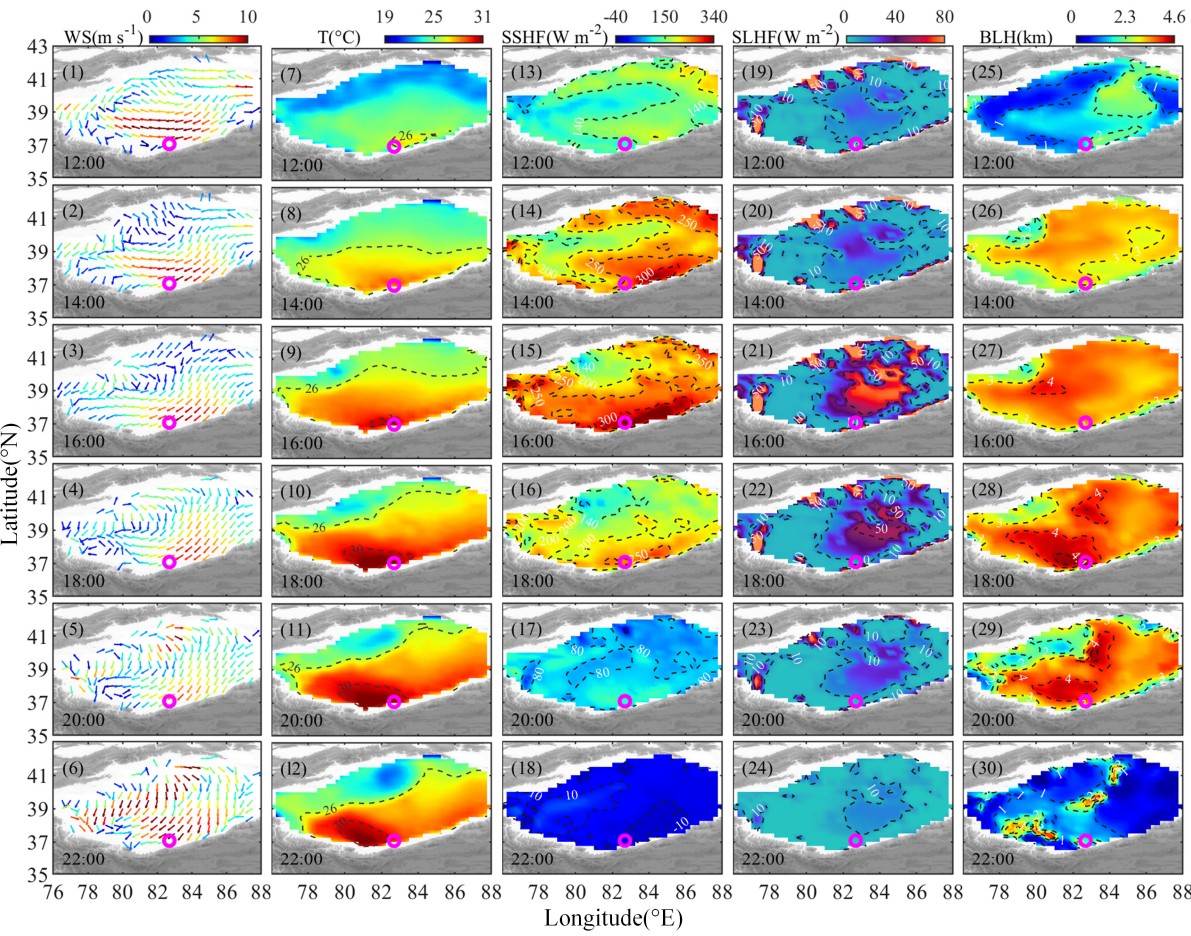

**Figure 6.** The distributions of ERA5 data over the Taklimakan Desert from 12:00 LT to 22:00 LT, on 6 June, 2022, local time (UTC+8). (1-6) wind vectors (850 hPa, 38 m above the ground). (7-12) air temperature (850 hPa, 38 m above the ground). (13-18) surface sensible heat flux. (19-24) surface latent heat flux. (25-30) boundary layer height. The purple circle represents the study site of MinFeng. The contours are denoted by dashed lines.

Fig. 7 illustrates meteorological data variation in the vertical section of the TD at the nearest study site of MinFeng. In Fig. 7(a-e), the wind vector is synthesized by ν and scaled ω (ω scaled by 10), which depicts the atmospheric vertical motion of the TD. The elevation of the study site is 1418 m. The Richardson number (Ri) can reflect the influence of vertical shear of horizontal wind on atmospheric stability and the state of atmospheric turbulence (Stull, 1988). There is a certain deviation in selecting different Ri values to identify the BLH (Guo et al., 2016). Generally, Ri values less than or equal to 0.25 are used to represent the turbulent motion state of the atmosphere, with a critical value of 0.25 often chosen to identify the BLH (Zhang et al., 2013). At 12:00 LT, the near-surface potential temperature on the left side is higher than that on the right side. Additionally, both the 20 % relative humidity contour line and the contour line representing the critical Ri value of 0.25 are maintained at a lower elevation.

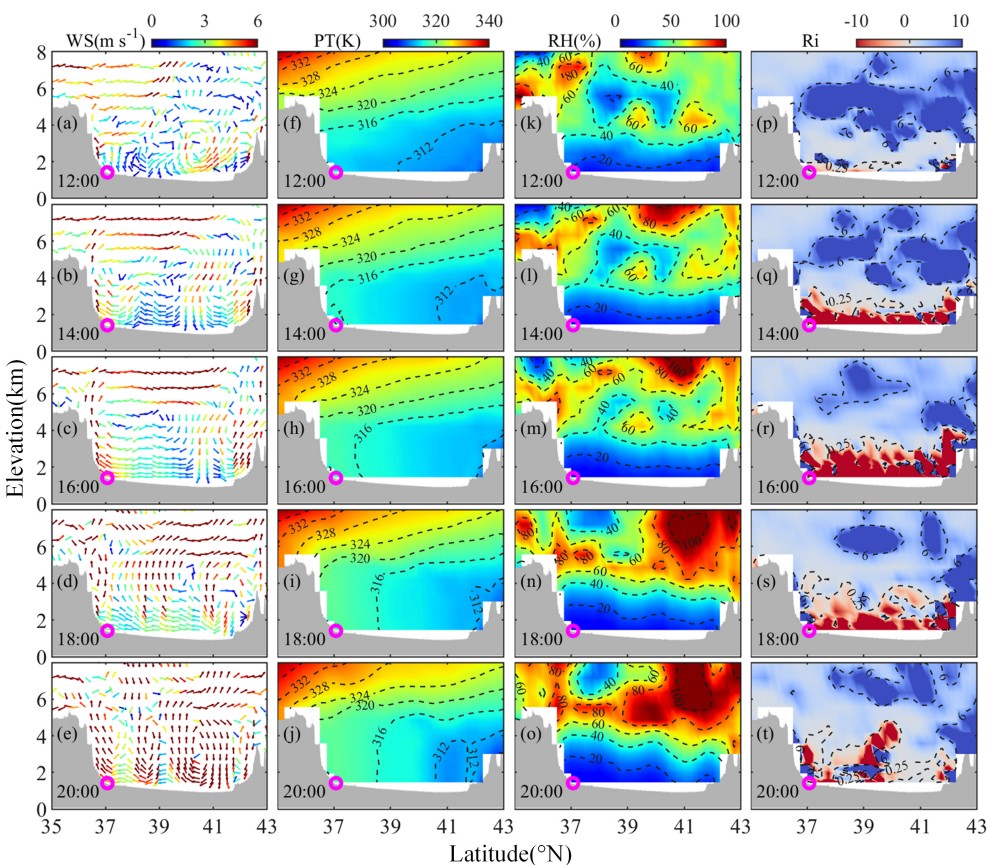

**Figure 7.** The meteorological elements in the vertical section of the Taklimakan Desert at the study site from 12:00 LT to 20:00 LT, on 6 June, 2022, local time (UTC+8). (a-e) wind vectors (synthesized by ν and scaled ω, ω scaled by 10). (f-g) potential temperature. (k-o) relative humidity. (p-t) Richardson number. The purple circle represents the study site of MinFeng. The leftmost of the study site is the Tibet Plateau.

At 14:00 LT~18:00 LT, within the elevation range of 1.4 km to 5 km, several atmospheric phenomena are observed. Firstly, the potential temperature on the north side of the TP remains nearly constant with elevation (Fig. 7(h-i), 316K). This indicates that the rate of temperature reduction during the rapid heating and expansion of the air exceeds the rate of dry adiabatic cooling,

forming an atmospheric superadiabatic expansion process. In this atmospheric state, a strong updraft can be formed (Fig. 7(b-c)), which makes the atmosphere in an unstable state. The obvious uplift process is visible in the 0.25 Ri contour in Fig. 7(q-s) and the 20% relative humidity contour in Fig. 7(l-n). This atmospheric superadiabatic expansion process significantly favors the diffusion and transport of dust aerosols and the development of boundary layer (Arnette et al., 1998; Nilsson et al., 2001).

This phenomenon also provides evidence that TKEDR is maintained within a large numerical range in Fig. 3c. Secondly, the topography and potential temperature of the south side of the desert basin are higher than those of the north side, creating an airflow from the northern side to the southern side (Fig. 7(b-c)). This allows dust aerosol can climb and transport along the Kunlun Mountains to the TP. Above the elevation of 5 km, a higher potential temperature area is distributed in the left TP, contributing to the formation of the TP-SHAP. There are two primary reasons for this: Firstly, the gradient of the potential

temperature of the TP is much higher than that of the desert basin, making the atmosphere unstable and easier to form atmospheric convective motion, and vertically suck dust aerosols around the TP (Jia et al., 2015; Wu et al., 2017; Feng et al., 2020). Secondly, at the same height, the potential temperature on the left side is higher than that on the right side, and the potential temperature slope on the left side is also larger, which can form a horizontal suction of dust aerosols over the desert. As depicted in Fig. 7(m-o), the relative humidity sinks from top to bottom above 4 km, while it gradually increases over the

study site. Similarly, Fig. A4n shows the rapid drop in temperature over the desert caused by the invasion of cold air. The cold air invaded the TD and intersected with the warm air over the desert, resulting in the formation of an upper-level cold front. As a consequence, the cold air sank to force the desert basin to produce a strong convective motion (Fig. 7d, Fig. 7e, also analyzed in Sect. 3). At about 18:00 LT, with the participation of the upper-level cold front, the height of deep CBL reached its peak (Fig. 3c, Fig. 6(28), Fig. 7s).

**5 Conclusion**

In this study, the CDWL data and ERA5 reanalysis data were used to comprehensively analyze the development process of a representative deep convective boundary layer on the southern edge of the Taklimakan Desert and the northern foot of the Tibet Plateau on 6 June, 2022. The results indicate that the formation of this deep convective boundary layer stems from the combined effects of multiple factors under complex terrain, including the Taklimakan Desert, slope terrain, the Kunlun

Mountains, and the Tibet Plateau. The primary factor is the low-level jet and inversion layer, which provide sufficient momentum, energy, and material prerequisites for the development of the atmospheric boundary layer. Furthermore, the thermal effect facilitates the formation of the deep convective boundary layer. The schematic diagram of the development of the low-level jet and the deep convective boundary layer is shown in Fig. 8.

The formation of low-level jet in the study site is attributed to the combined effect of multiple factors. The first is the pressure

gradient force and thermal difference between the north and south of the desert, which can form a background wind field from north to south. Secondly, the terrain blocking effect of the Tibet Plateau makes the northwest wind with lower wind speed deflect into the east wind with higher wind speed. Then, the terrain baroclinicity of the Tibet Plateau is easy for downhill

airflow to form at night, and when the downhill airflow is superimposed on the background wind field, the wind speed of the background wind field will be enhanced. Finally, the divergence structure of the upstream airflow and the convergence structure of the downstream airflow promote the development of the low-level jet.

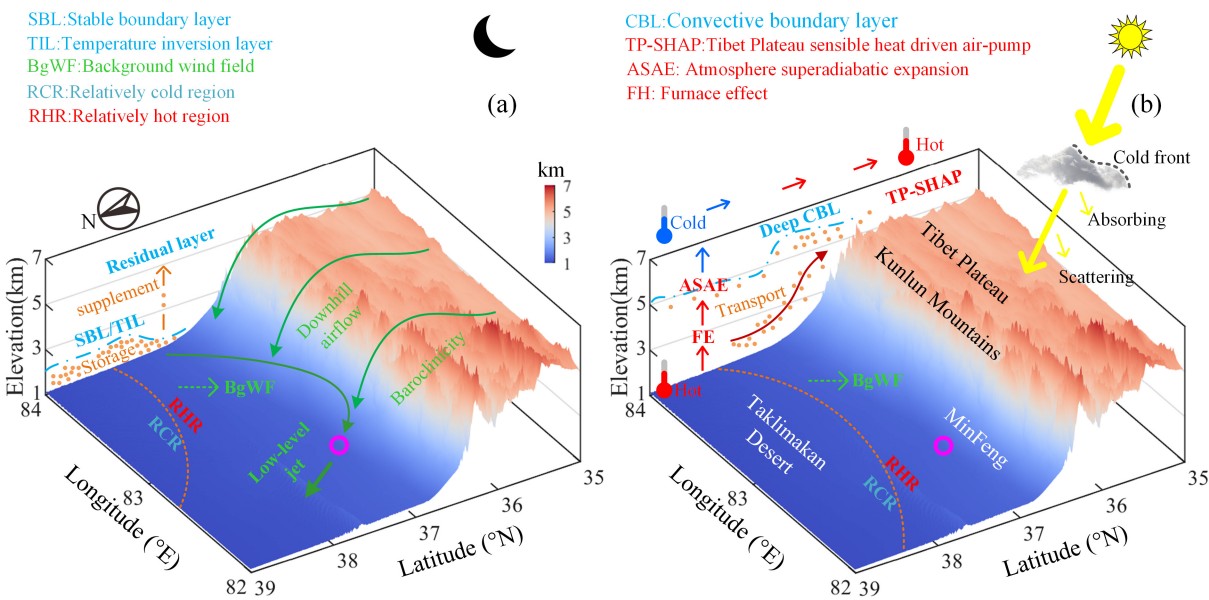

**Figure 8.** A schematic diagram of the development of low-level jet and deep convective boundary layer. (a) before sunrise. (b) before sunset.

The low-level jet and the inversion layer both play crucial roles in the development of the deep convective boundary layer. Firstly, the cold downhill wind contributes to the formation of the low-level jet, which can weaken the height and strength of the inversion layer, thereby reducing the energy demand for the breaking the inversion layer. Secondly, the low-level jet not only causes intermittent turbulence pulsations, but also inhibits the exchange of energy and material with the lower atmosphere, thus providing key material and energy supplements for the development of the deep convective boundary layer. Furthermore, the low-level jet can provide a dynamic basis for the subsequent development of the boundary layer. Finally, the existence of the inversion layer and the low-level jet can inhibit the internal turbulent motion in the lower atmosphere, causing dust aerosols to accumulate near the ground, thus providing material basis and storing sufficient momentum and energy for the subsequent development of the deep convective boundary layer.

The underlying surface of the desert itself has a strong heating effect on the atmosphere, and the corresponding thermal factors can catalyze the formation of the deep convective boundary layer. First of all, the accumulation of dust aerosols at low altitudes in the desert can form a furnace effect, rapidly transforming the material conditions for the development of the desert atmosphere boundary layer into thermal conditions, and promote the formation of the atmospheric superadiabatic expansion process. Secondly, the thermal effect of the Tibet Plateau sensible heat driven air-pump can suck up the atmosphere around the plateau and lift dust aerosols. Finally, the passage of a cold front can produce strong convective motion in the desert area.

Overall, the results reveal the formation process of a typical deep convective boundary layer in the Taklimakan Desert, and also reflect the process of land-atmosphere momentum, energy and material exchange and transport between the Taklimakan Desert and the Tibetan Plateau. However, the data range of CDWL only covers a point area in MinFeng, and lacks the observation results of the mountains along the northern side of the Tibet Plateau. The follow-up work will combine with multi-site observations of Raman lidar and weather radar to study the transmission characteristics of dust to the Tibet Plateau, as well as conduct long-term statistical analysis of the effects of extreme weather such as drought and dust storm on the boundary layer height.

*Data availability*

The ERA5 data sets are publicly available from ECMWF website at https://cds.climate.copernicus.eu. The two datasets used in ERA5 are "ERA5 hourly data on pressure levels from 1940 to present" and "ERA5 hourly data on single levels from 1940 to present". The CDWL data can be downloaded from https://figshare.com/articles/dataset/deep_CBL_lidar_datas/25434556 (Su et al., 2024a). The Fengyun-4A meteorological satellite data of China can be downloaded from http://www.nsmc.org.cn.

**Appendix A: The results of other observational data**

The cloud coverage over the Taklimakan Desert is shown in Fig. A1. The surface meteorological observation data on 6 June, 2022 (UTC + 8) are shown in Fig. A2. The CNR, horizontal wind speed, log10(TKEDR) and boundary layer height, vertical wind speed, wind direction during the field experiment in different seasons are shown in Fig. A3. Fig. A4 shows temperature variations at different altitudes over the Taklimakan Desert.

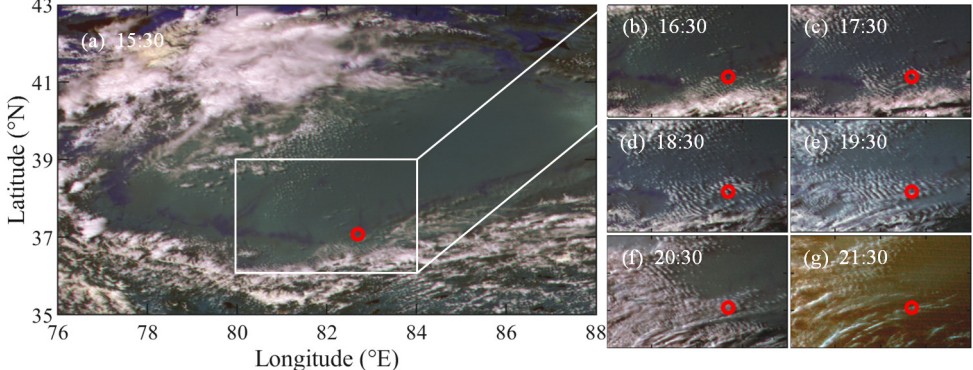

**Figure A1.** The cloud coverage over the Taklimakan Desert recorded by the Fengyun-4A meteorological satellite of China on 6 June, 2022 (UTC+8).

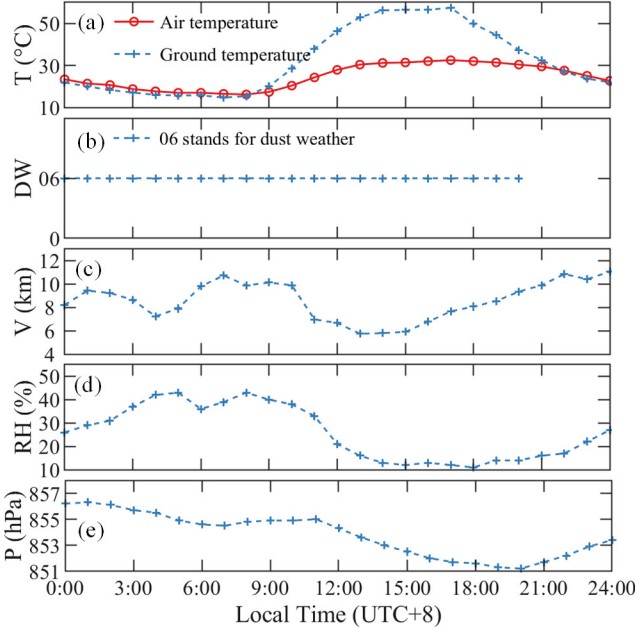

**Figure A2.** The surface meteorological observation data on 6 June, 2022, (UTC+8). (a) temperature. (b) dust weather. (c) horizontal visibility. (d) relative humidity. (e) local atmospheric pressure.

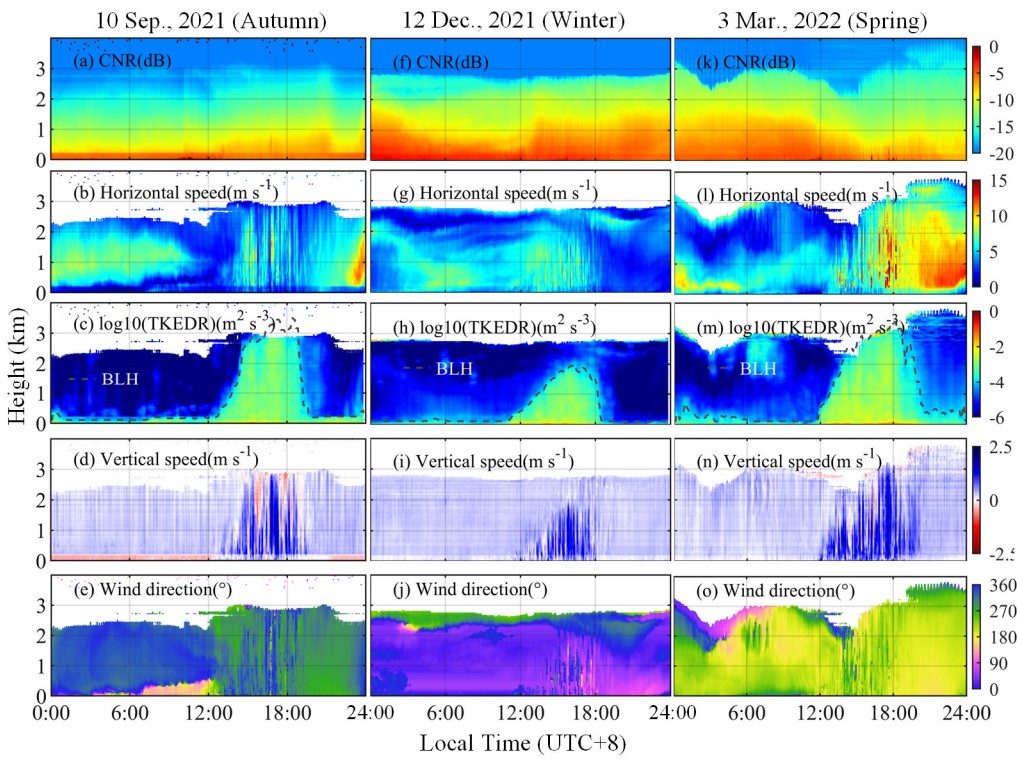

     **Figure A3.** The continuous observation results of CDWL in different seasons. (a-e) 10 Sep., 2021. (f-j) 12 Dec., 2021. (k-o) 3 Mar., 2022.

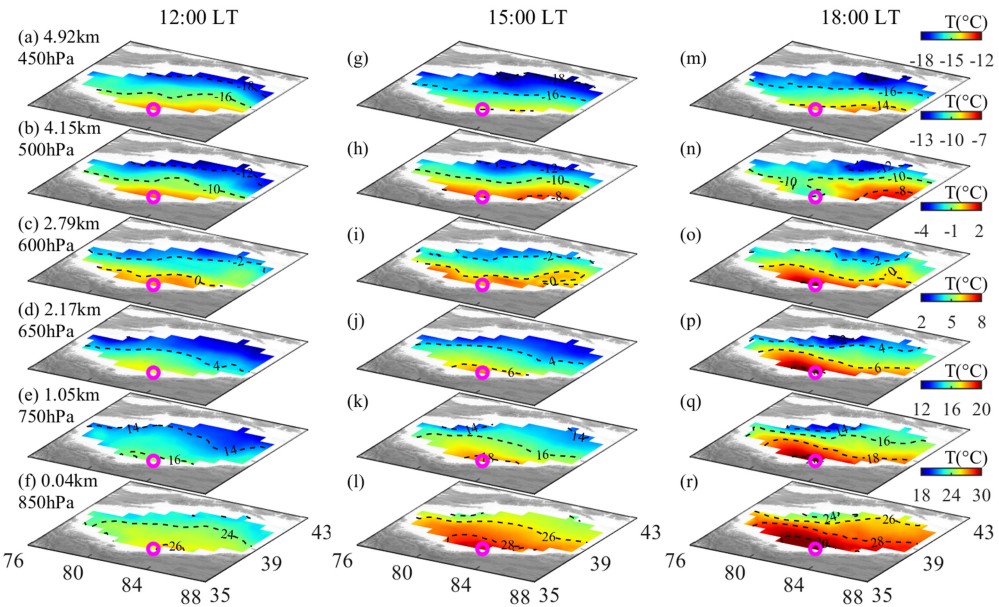

**Figure A4.** Temperature variations over the Taklimakan Desert during the period from 12:00 LT to 18:00 LT (UTC+8), observed at different atmospheric pressures or heights. The height represents the height from the ground at the study site. The purple circle represents the study site.

## 5   Appendix B: Methods

The CNR is obtained by the ratio of the signal area to the noise area of the power spectrum (Fujii and Fukuchi, 2005):

$$CNR = A_s/A_n \qquad (B1)$$

where $A_s$ is the signal area of the power spectrum, $A_n$ is the noise area of the power spectrum.

The line of sight velocity ($V_{los}$) of CDWL is given by the following formula:

10   $$V_{los} = \lambda f_i/2 \qquad (B2)$$

where $\lambda$ is the central wavelength of the emitted laser, $f_i$ is the Doppler frequency shift for aerosols.

The wind vector in the atmosphere can be expressed by $\vec{V}$:

$$\vec{V} = (u, v, w) \qquad (B3)$$

$u, v, w$ represent the north-south velocity, east-west velocity, and vertical velocity, respectively.

15   When using Velocity Azimuth Display (VAD) scanning mode, the direction vector $\vec{S}$ can be expressed as:

$$\vec{S} = (cos\theta cos\varphi, sin\theta cos\varphi, sin\varphi) \qquad (B4)$$

Where $\theta$ is the azimuth angle of the laser beam, and $\varphi$ is the elevation of the laser beam.

From formula B2, B3, and B4, it can be concluded that (Browning and Wexler, 1968):

$$V_{los} = \vec{V} \cdot \vec{S}_m \qquad (B5)$$

20   From formula B5, $u, v, w$ can be calculated. The horizontal wind direction is calculated as follows:

$$WD = arctan(u, v) \qquad\qquad\qquad\qquad\qquad\qquad\text{(B6)}$$

*Author contribution*

HX conceived, designed the study. LS and JY performed the lidar experiments. LS performed the analysis of lidar data and ERA5 data. XW and QH provide the field experiment site and the local meteorological data. LS carried out the analysis and 5  prepared the figures, with comments from other co-authors. LS, LC and JY wrote the manuscript. All authors read and approved the final manuscript.

*Competing interests.*

The authors declare that they have no conflict of interest.

*Acknowledgements.*

10  Thanks for the support of the Xinjiang Uygur Autonomous Region Meteorological Service and the Desert Meteorological Institute of China Meteorological Administration for this experiment and local meteorological data; The European Centre for Medium-Range Weather Forecasts for providing support with atmospheric reanalysis data.

*Financial support.*

This research has been supported by Strategic Priority Research Program of Chinese Academy of Sciences (grant no. 15  XDA22040601) and National Natural Science Foundation of China (grant no. 42030612).

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
