# Peer review of "Measurement report: The promotion of low-level jet and thermaleffect on development of deep convective boundary layer at the southern edge of the Taklimakan Desert"

_EGUsphere, 2024_

## Author Comment (AC1)

**Reviewer #1:**

Comments and Suggestions for Authors:

This manuscript reports lidar observations of an event with CBL as deep as over 5 km at the southern edge of the Taklimakan Desert. ERA5 reanalysis data was used to analyze the responsible mechanisms, as complemented by a conceptual diagram. The presented case is interesting and fits the scope of 'Measurement report' in ACP. I believe this work may be published upon addressing the issues given below.

Thanks a lot for your recognition of this work and for your carefully review. The manuscript has been greatly improved with your comments.

1. More discussions on the status of research on CBL should be given. Since this work aims to report an extreme CBL case, it is of necessity to give some background information on this point. For example, why the deep CBL is important? What are the known mechanisms contributing to deep CBL? Which knowledge gap that this study aims to address?

Some recent works have reported such deep CBL cases over this region. To what extent this work advances our understanding on the formation mechanism of deep CBL? This point will stand upon well addressing recent progresses regarding this topic.

**Response:** Thank you for pointing out these problems in the manuscript. We have added relevant background information to this manuscript.

Firstly, we have summarized the importance of studying deep CBL in the existing references, highlighting the necessity of conducting local studies on deep CBL. For example, "These studies also revealed that the deep CBL exerts an influence on the local pollutant transmission and diffusion, cloud formation processes, strong convective weather, rainfall, drought and so on", "However, the MinFeng station, which is located on the northern slope of the Tibet Plateau (TP) and has severe wind-sand activities (Yang et al., 2016; Xiao et al., 2008), was established in 2018 (Yang et al., 2020). On the one hand, there is a lack of sufficient study results of deep CBL, and the particularity of geographical locations (TD, slope terrain, Kunlun Mountains, TP) further complicates the formation mechanism of deep CBL. On the other hand, the deep CBL plays an important role in the regional circulation and weather, its study not only helps to reveal the mechanism of local dust emission and transport (Jia et al., 2015; Meng et al., 2019), but also promotes the understanding of the land-atmosphere interaction between the TD and the TP".

Secondly, we have introduced the existing mechanism of forming deep CBL in the Taklimakan Desert. For example, "in the hinterland of the TD, the intense surface heating is not the primary reason for the formation of deep CBL, whereas the presence of weak temperature inversion and near-neutral residual layer (RL) above the CBL are crucial (Zhang et al., 2022; Xu et al., 2018); The low-level jet (LLJ) can trigger significant air accumulation and dynamic convergence in the lower-level atmosphere, while the deep CBL is usually accompanied by the LLJ on the following night (Wang et al., 2019); The deep CBL enables the formation of clouds in the late afternoon, the formation of clouds will not only lead to a significant cooling of surface, but also make the momentum in the upper part of the boundary layer to transport downward and cause dust emissions (Zhang et al., 2024)".

Finally, in the conclusion part of the manuscript, we have summarized the key differences between the deep CBL formation mechanism presented in this study and those found in previous research. Previous studies have primarily concentrated on individual aspects, such as thermal or dynamic processes, to explain the formation of deep CBL. The formation of deep CBL in this

manuscript is a comprehensive effect of multiple factors under complex terrain (Taklimakan Desert, Tibet Plateau, slope terrain, Kunlun Mountains), including downhill airflow, low-level jet, inversion layer, the sensible heat driven air-pump from the Tibet Plateau, furnace effect, the atmospheric superadiabatic expansion process, cloud, and so on.

**Change:** P2L13-15. "These studies also revealed that the deep CBL exerts an influence on the local pollutant transmission and diffusion, cloud formation processes, strong convective weather, rainfall, drought and so on".

P2L16-30. "For example, in the hinterland of the TD, the intense surface heating is not the primary reason for the formation of deep CBL, whereas the presence of weak temperature inversion and near-neutral residual layer (RL) above the CBL are crucial (Zhang et al., 2022; Xu et al., 2018); The low-level jet (LLJ) can trigger significant air accumulation and dynamic convergence in the lower-level atmosphere, while the deep CBL is usually accompanied by the LLJ on the following night (Wang et al., 2019); The deep CBL enables the formation of clouds in the late afternoon, the formation of clouds will not only lead to a significant cooling of surface, but also make the momentum in the upper part of the boundary layer to transport downward and cause dust emissions (Zhang et al., 2024). Due to the fact that the TaZhong station is located in the center of the TD, and is equipped with the most comprehensive observation equipment, most studies of deep CBL are concentrated here. However, the MinFeng station, which is located on the northern slope of the Tibet Plateau (TP) and has severe wind-sand activities (Yang et al., 2016; Xiao et al., 2008), was established in 2018 (Yang et al., 2020). On the one hand, there is a lack of sufficient study results of deep CBL, and the particularity of geographical locations (TD, slope terrain, Kunlun Mountains, TP) further complicates the formation mechanism of deep CBL. On the other hand, the deep CBL plays an important role in the regional circulation and weather, its study not only helps to reveal the mechanism of local dust emission and transport (Jia et al., 2015; Meng et al., 2019), but also promotes the understanding of the land-atmosphere interaction between the TD and the TP".

P14L14-18. "The results indicate that the formation of this deep convective boundary layer stems from the combined effects of multiple factors under complex terrain, including the Taklimakan Desert, slope terrain, the Kunlun Mountains, and the Tibet Plateau. The primary factor is the low-level jet and inversion layer, which provide sufficient momentum, energy, and material prerequisites for the development of the atmospheric boundary layer. Furthermore, the thermal effect facilitates the formation of the deep convective boundary layer".

2. It is expected that the readers can reproduce your results after reading the Methods section, while the processing of lidar data is not adequately illustrated here.

**Response:** Thank you for your suggestion. We have introduced the data processing method of CDWL in the Appendix B of the manuscript, and the calculation of TKEDR is described in detail.

**Change:** P4L20-P5L9. "Based on this, the TKEDR threshold method can effectively estimate the BLH (Wang et al., 2021; Banakh et al., 2021).

The calculation formula of TKEDR is as follows (Banakh and Smalikho, 2018):

$$TKEDR = \left[\frac{\overline{D}_L(\varphi_l) - \overline{D}_L(\varphi_1)}{A(l\Delta y_k) - A(\Delta y_k)}\right]^{\frac{3}{2}} \qquad (1)$$

where $\overline{D}_L(\varphi_l)$ is azimuth structure function, $L$ is the serial number for the laser beam's line of sight. $\varphi_l = l\Delta\theta$, $\Delta\theta$ is the azimuth angle resolution, and $l=1,2,3…$. The $A(l\Delta y_k)$ is calculated theoretically for the Kolmogorov model of the two-dimensional turbulence spectrum (Banakh et al., 2017), $\Delta y_k$ is the

transverse dimension of the probed volume, and $k$ is the range gate number, $k$=1,2,3…. The error analysis for calculating TKEDR and BLH was conducted by Viktor A. Banakh (Banakh et al., 2017; Banakh et al., 2021).

In this experiment, the value of $l$ is set to 2, and the threshold of TKEDR is set to $10^{-4}$ m$^2$ s$^{-3}$. When the location is at the height of $H_n=\Delta R*N$ ($N$ is the index number of bins, and $\Delta R$ is radial spatial resolution), if all TKEDR values within the range $[\Delta R*(N+1), \Delta R*(N+5)]$ are less than the threshold, then $H_n$ is used as the BLH".

P18L6-P19L1. "The CNR is obtained by the ratio of the signal area to the noise area of the power spectrum (Fujii and Fukuchi, 2005):

$$CNR = A_s/A_n \qquad (B1)$$

where $A_s$ is the signal area of the power spectrum, $A_n$ is the noise area of the power spectrum.

The line of sight velocity ($V_{los}$) of CDWL is given by the following formula:

$$V_{los} = \lambda f_i/2 \qquad (B2)$$

where $\lambda$ is the central wavelength of the emitted laser, $f_i$ is the Doppler frequency shift for aerosols.

The wind vector in the atmosphere can be expressed by $\vec{V}$:

$$\vec{V} = (u, v, w) \qquad (B3)$$

$u$, $v$, $w$ represent the north-south velocity, east-west velocity, and vertical velocity, respectively.

When using Velocity Azimuth Display (VAD) scanning mode, the direction vector $\vec{S}$ can be expressed as:

$$\vec{S} = (cos\theta cos\varphi, sin\theta cos\varphi, sin\varphi) \qquad (B4)$$

Where $\theta$ is the azimuth angle of the laser beam, and $\varphi$ is the elevation of the laser beam.

From formula B2, B3, and B4, it can be concluded that (Browning and Wexler, 1968):

$$V_{los} = \vec{V} \cdot \vec{S}_m \qquad (B5)$$

From formula B5, $u$, $v$, $w$ can be calculated. The horizontal wind direction is calculated as follows:

$$WD = arctan\,(u, v) \qquad (B6)".$$

3. The impact of clouds is not well discussed. The presence of clouds may partially block the solar radiation to surface. The observed clouds seem to be rather thin in Figure 3, and the vertical air motions of clouds are similar to the dust layer below. The discussion around P7L11 is rather vague.

**Response:** Thank you for pointing out the shortcomings in the manuscript, we have made a lot of improvements to the manuscript. Firstly, in Figure 3(c), the height of the cloud base has been marked with a purple dotted line based on the HWCT (Haar wavelet covariance transform) method. Secondly, in Appendix A3, the cloud coverage of the Taklimakan Desert was drawn using the Fengyun-4A meteorological satellite. Finally, we discussed the impact of clouds from four aspects in the manuscript.

The first is the distribution of clouds. For example, "Before 18:00 LT, the study site was covered by scattered clouds, and after 18:00 LT, the cloud completely covered the study site".

The second is the radiative cooling effect of clouds. For example, "The presence of clouds can greatly weaken the solar radiation reaching the surface, causing the surface temperature to decrease rapidly (Fig. 3a)", "When the cloud completely covered the study site, the surface radiation further cooled the near-surface air, greatly weakening the atmospheric turbulence intensity and significantly reducing the CBL height before sunset, and the ground-air temperature difference changed to -0.6 °C at 22:00 LT".

Then the cloud caused the change of atmospheric turbulence intensity. For example, "and gradually weakening the turbulence intensity of the atmosphere (Fig. 3c)", "At 17:00 LT~20:00 LT, the

atmospheric turbulence remained active during the initial stages of cloud formation".

Finally, the formation of upper-level cold front was promoted. For example, "the cold clouds moved towards the warm air mass over the desert, promoting the formation of an upper-level cold front and causing strong convective motion in the lower atmosphere (Fig. 3d), and the height of deep CBL reached its peak at 18:00 LT (Fig. 3c)".

In the aspects of the vertical movement of clouds and dust. On the one hand, due to the attenuation of the lidar signal by the cloud layer, most of the detected clouds are near the cloud base. On the other hand, during the early stages of cloud formation, the movement of cold clouds towards the warm air over the desert triggers intense convective motion in the lower atmosphere, resulting in similar vertical movements between the clouds and dust particles. However, after 20:00 LT, the vertical motion of the upper layer, which is higher than 5 km, is significantly greater than that of the lower layer.

**Change:** P7L19-29. "the study site began to be covered by clouds (Fig. 3c, Fig. A3). Before 18:00 LT, the study site was covered by scattered clouds, and after 18:00 LT, the cloud completely covered the study site. The presence of clouds can greatly weaken the solar radiation reaching the surface, causing the surface temperature to decrease rapidly (Fig. 3a), and gradually weakening the turbulence intensity of the atmosphere (Fig. 3c). At 17:00 LT~20:00 LT, the atmospheric turbulence remained active during the initial stages of cloud formation. On the one hand, as the surface temperature was still much higher than the atmospheric temperature, and the heat preservation effect of dust on the atmosphere continued to provide energy for the upper atmosphere. On the other hand, the cold clouds moved towards the warm air mass over the desert, promoting the formation of an upper-level cold front and causing strong convective motion in the lower atmosphere (Fig. 3d), and the height of deep CBL reached its peak at 18:00 LT (Fig. 3c). When the cloud completely covered the study site, the surface radiation further cooled the near-surface air, greatly weakening the atmospheric turbulence intensity and significantly reducing the CBL height before sunset, and the ground-air temperature difference changed to -0.6 °C at 22:00 LT".

P7L34. "The cloud coverage over the Taklimakan Desert is shown in Fig. A3".

P8L4-5. "and the cloud base height is marked with a purple dotted line".

P14L5-10. "As shown in Fig. 7(m-o), the relative humidity sinks from top to bottom above 4 km, while over the study site it gradually increases. Fig. A4n, on the other hand, shows the rapid drop in temperature over the desert caused by the invasion of cold air. The cold air invaded the TD and intersected with the warm air over the desert to form an upper-level cold front, and the cold air sank to force the desert basin to produce a strong convective motion (Fig. 7d, Fig. 7e, also analyzed in Sect. 3). At about 18:00 LT, with the participation of the upper-level cold front, the height of deep CBL reached its peak (Fig. 3c, Fig. 6(28), Fig. 7s)".

P16L9. "The cloud coverage over the Taklimakan Desert is shown in Fig. A3".

P17L5-6. "Figure A3. The cloud coverage over the Taklimakan Desert recorded by the Fengyun-4A Meteorological Satellite of China on 6 June, 2022 (UTC+8).".

[Figure]

**Figure A3.** The cloud coverage over the Taklimakan Desert recorded by the Fengyun-4A Meteorological Satellite of China on 6 June, 2022 (UTC+8).

4. Causality issue. One of the major conclusions is that LLJ plays an important role in forming deep CBL. I do see that the study site was in the LLJ region, however, I did not find strong evidence showing how LLJ contributes to the formation of deep CBL in section 4.1. At the very least, one may be convinced if you show the well correlated time series of LLJ and BLH. Therefore, I suggest the authors to reorganize this section, and discuss this point more concisely and logically.

**Response:** Thanks for your kind reminder. We have added some descriptions of the role of LLJ in the development of deep CBL in section 4.1. For example, "In summary, the LLJ formed a water vapor convergence area in front of the study site and maintained the temperature within a relatively high range (16°C isotherm), which enhanced the potential instability (Fig. 4(g-i)), thereby strengthening the convective potential of the atmosphere and providing the necessary energy and water vapor conditions for the subsequent development of the boundary layer", "At the study site, the atmospheric temperature change caused by the LLJ promoted the formation of the IL, and the most obvious IL phenomenon was observed at 8:00 LT. The IL can weaken the convective motion of atmosphere, resulting in the boundary layer height near the study site being constrained within 0.25 km, thereby limiting the diffusion and mixing of dust pollutants, and serving as a source of dust material for further boundary layer development".

However, in section 3, the importance of LLJ to the formation of deep CBL has been fully demonstrated through lidar data. At the same time, the corresponding relationship between BLH and LLJ was also given in time series (Fig.3b and Fig.3c). Therefore, in Section 4.1, the causes and development of the LLJ in this area were mainly discussed, and the correctness of the CDWL observation was also confirmed. In addition, it is worth noting that the spatial resolution of ERA5 is 0.25°×0.25°, and the temporal resolution is 1 hour, which makes it difficult to study the spatiotemporal behavior of the CBL of this day at the study site.

**Change:** P9L10. "As seen in the wind vector subgraphs".

P9L10-11. "thus confirming the validity of observing LLJ using CDWL".

P10L8-11. "In summary, the LLJ formed a water vapor convergence area in front of the study site and maintained the temperature within a relatively high range (16°C isotherm), which enhanced the potential instability (Fig. 4(g-i)), thereby strengthening the convective potential of the atmosphere and providing the necessary energy and water vapor conditions for the subsequent development of the boundary layer".

P11L9-13. "At the study site, the atmospheric temperature change caused by the LLJ promoted the

formation of the IL, and the most obvious IL phenomenon was observed at 8:00 LT. The IL can weaken the convective motion of atmosphere, resulting in the boundary layer height near the study site being constrained within 0.25 km, thereby limiting the diffusion and mixing of dust pollutants, and serving as a source of dust material for further boundary layer development".

5. From Fig.8b, it seems that the cold front is tangentially related to the deep CBL. Looking back to the discussion at P13L23, I am not sure how the processes could be interpreted from the figure given.

**Response:** Thank you for your question, we further revised the manuscript. We have added Fig. A4 in Appendix A to show the temperature variations at different heights. Firstly, from Fig. A4n, it can be found that the cold air moving with the cloud greatly cooled the upper part of the study site. Secondly, in Fig. 7(m-o), the relative humidity sinks from top to bottom above 4 km. Thirdly, in Fig. 3d, Fig.7d, and Fig.7e, the cold air mass actively moves towards the warm air mass over the desert, thus forcing the desert basin to produce a strong convective motion. Finally, with the participation of the upper-level cold front, the height of the boundary layer reached its peak at approximately 18:00 LT.

**Change:** P7L25-27. "On the other hand, the cold clouds moved towards the warm air mass over the desert, promoting the formation of an upper-level cold front and causing strong convective motion in the lower atmosphere (Fig. 3d), and the height of deep CBL reached peak at 18:00 LT (Fig. 3c)."

P7L34. "The cloud coverage over the Taklimakan Desert is shown in Fig. A3".

P14L5-10. "As shown in Fig. 7(m-o), the relative humidity sinks from top to bottom above 4 km, while over the study site it gradually increases. Fig. A4n, on the other hand, shows the rapid drop in temperature over the desert caused by the invasion of cold air. The cold air invaded the TD and intersected with the warm air over the desert to form an upper-level cold front, and the cold air sank to force the desert basin to produce a strong convective motion (Fig. 7d, Fig. 7e, also analyzed in Sect. 3). At about 18:00 LT, with the participation of the upper-level cold front, the height of deep CBL reached its peak (Fig. 3c, Fig. 6(28), Fig. 7s).".

P16L9-10. "Fig. A4 shows temperature variations at different altitudes over the Taklimakan Desert".

P18L2-4. "Figure A4. Temperature variations over the Taklimakan Desert during the period from 12:00 LT to 18:00 LT (UTC+8), observed at different atmospheric pressures or heights. The height represents the height from the ground at the study site. The purple circle represents the study site.".

[Figure]

**Figure A4**. Temperature variations over the Taklimakan Desert during the period from 12:00 LT to 18:00 LT (UTC+8), observed at different atmospheric pressures or heights. The height represents the height from the ground at the study site. The purple circle represents the study site.

6. ACP usually has very good production team, but I would suggest to find a native speaker to improve the language issues. Some awkward expressions may be revised.

**Response:** Thank you for your kind reminder, we have carefully revised the English language. All changes made in the revised manuscript are marked in blue.

**Change:** P1L17-18. "During the stage of LLJ prior to the formation of the deep CBL, the LLJ had made sufficient preparations for the development of the deep CBL in terms of momentum, energy, and material".

P1L20. "temperature inversion layer".

P1L22. "During the stage of thermal effects, During the stage of thermal effects, the sensible heat driven air-pump from the Tibet Plateau".

P3L1. "In the detection of the BLH".

P3L2-6. "These characteristics of CDWL enable it to obtain the air flow conditions of the atmosphere from the calculated wind field information, monitor the change of the BLH in real time and more accurately, and help to understand the diffusion and retention of dust pollutants. Overall, the CDWL can be used for long-term continuous and stable detection in desert areas, and it is one of the effective ways to estimate the BLH in the desert".

…

---

## Author Comment (AC2)

**Reviewer #2:**

Comments and Suggestions for Authors:

1. This paper investigates the formation of a deep convective boundary layer on June 6, 2022, at the southern edge of the Taklimakan Desert and the northern foot of the Tibetan Plateau, using coherent Doppler wind lidar (CDWL) and ERA5 reanalysis data. The study reveals the critical role of low-level jets and thermal effects in the development of the boundary layer, providing significant scientific insights into the transport of dust aerosols and the thermodynamic processes of the atmospheric boundary layer. Overall, this paper is well-written, scientifically rigorous, and presents strong conclusions with high publication value. I recommend it for publication after some additional details and further analysis are addressed.

    **Response:** Thank you for your recognition of this work. The manuscript has been greatly improved with your comments.

2. The paper does not provide a detailed explanation of how the boundary layer height is estimated using Equation (1) in the methods section. It is recommended that the authors provide a more detailed explanation of how the boundary layer height is derived.

**Response:** Thank you very much for this professional suggestion. We have made a more detailed for the calculation process of BLH in Sect. 2. and we have introduced the data processing method of CDWL in the Appendix B of the manuscript.

**Change:** P4L20-P5L9. "Based on this, the TKEDR threshold method can effectively estimate the BLH (Wang et al., 2021; Banakh et al., 2021).

The calculation formula of TKEDR is as follows (Banakh and Smalikho, 2018):

$$TKEDR = [\frac{\overline{D}_L(\varphi_l) - \overline{D}_L(\varphi_1)}{A(l\Delta y_k) - A(\Delta y_k)}]^{\frac{3}{2}} \tag{1}$$

where $\overline{D}_L(\varphi_l)$ is azimuth structure function. $L$ is the serial number for the laser beam's line of sight. $\varphi_l = l\Delta\theta$, $\Delta\theta$ is the azimuth angle resolution, and $l$=1,2,3,4…. The $A(l\Delta y_k)$ is calculated theoretically for the Kolmogorov model of the two-dimensional turbulence spectrum (Banakh et al., 2017), and $\Delta y_k$ is the transverse dimension of the probed volume, and $k$ is the range gate number, $k$=1,2,3….. The error analysis for calculating TKEDR and MLH was conducted by Viktor A. Banakh (Banakh et al., 2017; Banakh et al., 2021).

In this experiment, the value of $l$ is set to 2, and the threshold of TKEDR is set to $10^{-4}$ $m^2$ $s^{-3}$. When the location is at the height of $H_n = \Delta R * N$ ($N$ is the index number of bins, and $\Delta R$ is radial spatial resolution), if all TKEDR values within the range $[\Delta R*(N+1), \Delta R*(N+5)]$ are less than the threshold, then $H_n$ is used as the BLH".

P5L26. "When the CNR value is below -17 dB, the calculated products are eliminated in the preprocessing".

P18L6-P19L1. "The CNR is obtained by the ratio of the signal area to the noise area of the power spectrum (Fujii and Fukuchi, 2005):

$$CNR = A_s/A_n \tag{B1}$$

where $A_s$ is the signal area of the power spectrum, $A_n$ is the noise area of the power spectrum.

The line of sight velocity ($V_{los}$) of CDWL is given by the following formula:

$$V_{los} = \lambda f_i/2 \tag{B2}$$

where $\lambda$ is the central wavelength of the emitted laser, $f_i$ is the Doppler frequency shift for aerosols.

The wind vector in the atmosphere can be expressed by $\vec{V}$:

$$\vec{V} = (u, v, w) \tag{B3}$$

$u$, $v$, $w$ represent the north-south velocity, east-west velocity, and vertical velocity, respectively.

When using Velocity Azimuth Display (VAD) scanning mode, the direction vector $\vec{S}$ can be expressed as:

$$\vec{S} = (cos\theta cos\varphi, sin\theta cos\varphi, sin\varphi) \tag{B4}$$

Where $\theta$ is the azimuth angle of the laser beam, and $\varphi$ is the elevation of the laser beam.

From formula B2, B3, and B4, it can be concluded that (Browning and Wexler, 1968):

$$V_{los} = \vec{V} \cdot \vec{S}_m \tag{B5}$$

From formula B5, $u$, $v$, $w$ can be calculated. The horizontal wind direction is calculated as follows:

$$WD = arctan\,(u, v) \tag{B6}$$".

3. The paper mentions the impact of the downhill airflow on the low-level jet and the inversion layer but lacks specific observational data and analysis. On page 5, line 21, the paper states: "At 0:00 LT~6:00 LT, before the formation of the LLJ, the downhill airflow blowing from the TP to the desert was superimposed on the desert background wind field." However, Figure 3e shows that the region is dominated by north and northeast winds, suggesting that the descending airflow is likely not from the Tibetan Plateau. Additionally, based on Figure 4, I believe that the local temperature drop is more likely due to the transport of cooler air from the upstream of the jet stream. It is recommended that the authors provide more information on the downhill airflow to make it easier for readers to understand.

**Response:** Thank you for your kind reminder. The geographical location of the study site is shown in the following figure, and the study site located at the northern foot of the Tibetan Plateau. The northeast part of the study site is blocked by the Tibet Plateau. In this manuscript, first of all, 0° represents the wind blows to the north. From 0:00 LT to 6:00 LT, the wind direction above 3 km is approximately distributed between 225° and 295° (Fig. 3e), and there is a clear sinking airflow in the horizontal wind speed figure (Fig. 3b). Secondly, it can be seen from the wind vectors in Fig. 4(a-c) that the upstream wind speed of the study site is very small, but after reaching the study site, it changes to a strong easterly wind, and the potential becomes lower (Fig. 4h), which can demonstrate the superposition effect of the downhill airflow. Finally, combined with the geographical location map, it can be seen that the superposition effect of the downhill airflow on the Tibetan Plateau induces the formation of LLJ.

It's worth noting that the cooling at the study site is also affected by cold air transported by the upstream of the LLJ, we have also added some discussion on this in the manuscript.

**Change:** P6L6-8, "It can be clearly found in Fig. 3b that the horizontal wind speed has subsided from 4 km to 2 km, and the wind is the downhill wind blowing from the TP to the desert (Fig .3e, >3 km, 225°-295°). When the downhill airflow was superimposed on the desert background wind field".

P6L14-15, "with the cold downhill airflow and the upstream cold airflow traveled to the desert basin where the study site is located".

[Figure]

**Figure C1.** The 3D topographic map of the study area, and the purple circle represent the study site of MinFeng

4. The paper mentions that dust aerosols have an impact on surface and atmospheric temperatures but lacks specific quantitative analysis. Providing the diurnal variation of dust concentration could more intuitively show the impact of dust concentration changes on the boundary layer.

**Response:** Thank you very much for this professional suggestion. The MinFeng station was built in 2018. At present, there is still a lack of data reflecting dust concentration such as PM10 on the day of the experiment. However, the CNR (Fig. 3a) can be used as indirect indicators of dust concentration, and they reflect the distribution of dust concentration. For example, "the atmosphere exhibited a stratified state (Blackadar, 1957), so that the high concentration of CNR values was distributed below 1 km, which can provide a material basis for boundary layer development", "it can be seen that the values of TKEDR were consistently maintained at a high level, indicating a significant enhancement in the vertical transport capacity of the atmosphere (Wang et al., 2020). Consequently, the underlying CNR value below 1 km in Fig. 3a increased significantly".

**Change:** P5L25-26. "The CNR can be used as an indicator of aerosol concentration (Pea et al., 2013)".
P6L27-28. "the atmosphere exhibited a stratified state (Blackadar, 1957), so that the high concentration of CNR values was distributed below 1 km, which can provide a material basis for boundary layer development".
P7L1-3. "it can be seen that the values of TKEDR were consistently maintained at a high level, indicating a significant enhancement in the vertical transport capacity of the atmosphere (Wang et al., 2020). Consequently, the underlying CNR value below 1 km in Fig. 3a increased significantly".

5. An interesting phenomenon can be observed in Figure 2, where higher boundary layers occur more frequently in spring and summer. Therefore, are the effects of the low-level jet and thermal effects on the deep convective boundary layer proposed in this study seasonal? We look forward to the authors addressing this question in the discussion section.

**Response:** It is my pleasure to discuss this with you. The following figure shows the wind frequency rose during a 2-year period at the study site. In spring, firstly, the temperature of the underlying surface rises rapidly after being heated by solar radiation, which accelerates the freeze-thaw alternation of the soil. Secondly, the east-west airflows converge here, making it the most frequent location for dust storm disaster in China, and the boundary layer also develops vigorously. Finally, according to the principle of inertial oscillation, the LLJ of east direction prevails in spring. In summer, the strong

westerly wind dominates the study site, and the characteristics of the dust underlying surface have the most significant heating effect on the atmosphere, resulting in strong convection and turbulent activity during the daytime, which is conducive to the formation of deep CBL. In terms of LLJ formation, the first is that there is easy to form an inversion layer due to the rapid cooling of the dust underlying surface, so that the lower layer of the boundary layer is in a stable stratification and is conducive to the excitation of the inertial oscillation mechanism. Secondly, there is a large temperature difference between daytime and nighttime, as well as the different cooling amplitude on the slope terrain, which makes it easier for downhill airflow to form, thereby promoting the development of LLJ.

[Figure]

**Figure C2.** The rose diagram of wind frequency on the surface of study site from July 2021 to July 2023. The wind speed of 3.5 m/s can be used as the critical wind speed of blowing dust.

6. Why did the authors use sea level air pressure in the study? Sea level pressure is absent in TP and Taklimakan Desert due to the presence of terrain. Is it reasonable to use the difference in sea level pressure to analyze the formation of downhill airflow?

**Response:** Thank you for your question. Due to the high altitude and thin air of the Tibet Plateau, the local surface atmospheric pressure is generally lower than that of the desert basins. Sea level pressure refers to the atmospheric pressure per unit area calculated from the height of the sea level. Usually, in meteorological applications, atmospheric pressure values at different altitudes are typically converted to pressure values above sea level in order to establish standards and facilitate comparisons. Therefore, we have adopted sea level air pressure in this manuscript.

7. Other specific comments.

**Response:** Thank you for your kind reminding, we have carefully revised the manuscript. The clarity of the picture has also been improved. All changes made in the revised manuscript are marked in blue.

**Change:** P1L17-18. "During the stage of LLJ prior to the formation of the deep CBL, the LLJ had made sufficient preparations for the development of the deep CBL in terms of momentum, energy, and material."

P1L20. "temperature inversion layer".

P1L23. "During the stage of thermal effects, the sensible heat driven air-pump from the Tibet Plateau".

P2L10-15. "These studies also revealed that the deep CBL exerts an influence on the local pollutant transmission and diffusion, cloud formation processes, strong convective weather, rainfall, drought and so on".

P2L16-30. "For example, in the hinterland of the TD, the intense surface heating is not the primary

reason for the formation of deep CBL, whereas the presence of weak temperature inversion and near-neutral residual layer (RL) above the CBL are crucial (Zhang et al., 2022; Xu et al., 2018); The low-level jet (LLJ) can trigger significant air accumulation and dynamic convergence in the lower-level atmosphere, while the deep CBL is usually accompanied by the LLJ on the following night (Wang et al., 2019); The deep CBL enables the formation of clouds in the late afternoon, the formation of clouds will not only lead to a significant cooling of surface, but also make the momentum in the upper part of the boundary layer to transport downward and cause dust emissions (Zhang et al., 2024). Due to the fact that the TaZhong station is located in the center of the TD, and is equipped with the most comprehensive observation equipment, most studies of deep CBL are concentrated here. However, the MinFeng station, which is located on the northern slope of the Tibet Plateau (TP) and has severe wind-sand activities (Yang et al., 2016; Xiao et al., 2008), was established in 2018 (Yang et al., 2020). On the one hand, there is a lack of sufficient study results of deep CBL, and the particularity of geographical locations (TD, slope terrain, Kunlun Mountains, TP) further complicates the formation mechanism of deep CBL. On the other hand, the deep CBL plays an important role in the regional circulation and weather, its study not only helps to reveal the mechanism of local dust emission and transport (Jia et al., 2015; Meng et al., 2019), but also promotes the understanding of the land-atmosphere interaction between the TD and the TP.".

…

---

## Author Response (AR2)

* * *
Manuscript ID: egusphere-2024-1010

Title: Measurement report: The promotion of low-level jet and thermal-effect on development of deep convective boundary layer at the southern edge of the Taklimakan Desert

Corresponding authors: Haiyun Xia; hsia@ustc.edu.cn; University of Science and Technology of China
* * *
Dear Editors

  On behalf of the co-authors, thank you for giving us an opportunity to revise the manuscript. We appreciate the great efforts and constructive comments from the reviewers, which improve the quality of the manuscript significantly. We have revised the manuscript carefully according to the reviewers' comments and suggestions. Our point-by-point responses are appended below. All changes made in the revised manuscript are marked in light blue. Attached please find the revised version of the manuscript, which we would like to submit for your kind consideration. We are looking forward to hearing from you!

Best regards!
Sincerely yours,
Haiyun Xia
School of Earth and Space Science
University of Science and Technology of China.
Hefei, Anhui, CHINA, 230026.
* * *
**Reviewer #1:**

Comments and Suggestions for Authors:

1. Most of my concerns have been well addressed. However, the research question is still yet established. The introduction left me the impression that you have the observations at the Minfeng station, and you just report the observations. Please elaborate why the deep convective boundary layer over Minfeng station is important instead of saying that you have the unique observations.

**Response:** Thank you for pointing out these problems in the manuscript. The manuscript has been greatly improved with your comments. We illustrated the importance of the study of deep convective boundary layer at Minfeng station from three aspects.

First of all, the Taklimakan Desert is an important dust source area in China. Under the influence of the deep convective boundary layer and the driving force of the northern slope of the Tibet Plateau, dust aerosol in the study site can rise to higher than 7 km (Meng et al., 2019), which affects regional and even global precipitation, cloud cover, and material circulation during long-distance transportation (Ge et al., 2014; Huang et al., 2014).

Secondly, the formation of the deep convective boundary layer is often accompanied by strong mixing of atmospheric pollutants in the vertical direction. At the study site, the annual average number of days with dust weather is 113.5 (Yang et al., 2016), and the number of days with a boundary layer height exceeding 4 km in summer is more than that observed at other major weather stations in the Taklimakan Desert (Wang et al., 2019). Studying the deep convective boundary layer is helpful for understanding the formation and evolution of dust pollution weather and contributes to the management of the ecological environment.

Thirdly, in the study area, special meteorological phenomena such as drought, severe convective weather, dust storms, gales, low-level jets, wind shear, and others often occur concomitantly with the development of the deep convective boundary layer (Su et al., 2024; Wang et al., 2016; Ge et al., 2016). Therefore, studying the deep convective boundary layer holds significant research importance for understanding these special meteorological phenomena.

**Change:** P2L24-P3L2. "The unique geographical location of the study site (TD, slope terrain, Kunlun Mountains, TP) makes the formation mechanism of the deep CBL not only complex but also highly significant. For example, within the study area, special meteorological phenomena such as drought, severe convective weather, dust storms, gales, low-level jets, wind shear, and others frequently occur concomitantly with the development of the deep CBL (Su et al., 2024b; Wang et al., 2016; Ge et al., 2016). The annual average number of days with dust weather is 113.5 (Yang et al., 2016), and during summer, the number of days with a BLH exceeding 4 km surpasses that observed at other major weather stations within the TD (Wang et al., 2019). Investigating the deep CBL is instrumental in comprehending the formation and evolution of dust pollution weather and contributes to the management of the ecological environment. Furthermore, under the combined influence of the deep CBL and the driving force emanating from the northern slope of the TP, dust aerosols within the study site have the capability to ascend to heights exceeding 7 km (Meng et al., 2019), ultimately impacting regional and potentially even global precipitation patterns, cloud cover, and material circulation during their long-distance transportation (Ge et al., 2014; Huang et al., 2014)".

**References:**

Ge, J., Liu, H., Huang, J., and Fu, Q.: Taklimakan Desert nocturnal low-level jet: climatology and dust activity, Atmos Chem Phys, 16, 7773-7783, 10.5194/ACP-16-7773-2016, 2016.

Ge, J., Huang, J., Xu, C., Qi, Y., and Liu, H.: Characteristics of Taklimakan dust emission and distribution: A satellite and reanalysis field perspective, Journal of Geophysical Research: Atmospheres, 119, 11,772-711,783, 10.1002/2014JD022280, 2014.

Huang, J., Wang, T., Wang, W., Li, Z., and Yan, H.: Climate effects of dust aerosols over East Asian arid and semiarid regions, Journal of Geophysical Research: Atmospheres, 119, 11,398 - 311,416, 10.1002/2014JD021796, 2014.

Meng, L., Yang, X.-h., Zhao, T., He, Q., Lu, H., Mamtimin, A., Huo, W., Yang, F., and Liu, C.: Modeling study on three-dimensional distribution of dust aerosols during a dust storm over the Tarim Basin, Northwest China, Atmospheric Research, 218, 285-295, 10.1016/J.ATMOSRES.2018.12.006, 2019.

Su, L., Xia, H., Yuan, J., Wang, Y., Maituerdi, A., and He, Q.: Study on Daytime Atmospheric Mixing Layer Height Based on 2-Year Coherent Doppler Wind Lidar Observations at the Southern Edge of the Taklimakan Desert, Remote Sensing, 16, 3005, 10.3390/rs16163005, 2024.

Wang, M., Wei, W., He, Q., Yang, Y., Fan, L., and Zhang, J.: Summer atmospheric boundary layer structure in the hinterland of Taklimakan Desert, China, Journal of Arid Land, 8, 846-860, 10.1007/s40333-016-0054-3, 2016.

Wang, M., Xu, X., Xu, H., Lenschow, D. H., Zhou, M., Zhang, J., and Wang, Y.: Features of the deep atmospheric boundary layer over the Taklimakan Desert in the summertime and its influence on regional circulation, Journal of Geophysical Research: Atmospheres, 124, 12755-12772, 10.1029/2019JD030714, 2019.

Yang, X., Shen, S., Yang, F., He, Q., Ali, M., Huo, W., and Liu, X.: Spatial and temporal variations of blowing dust events in the Taklimakan Desert, Theoretical and Applied Climatology, 125, 669-677, 10.1007/s00704-015-1537-4, 2016.

2. Also, please check the revised manuscript. Seems to be too many 'on the other hand' ... As I suggested already, please find a native speaker to revise the language.

**Response:** Thank you for your kind reminder. We have made great efforts to improve the smoothness and grammar of the sentences in the manuscript.

**Change:** P1L17-18. "During the stage of LLJ preceding the formation of the deep CBL, the LLJ had adequately prepared the conditions for the development of the deep CBL in terms of momentum, energy, and material".

P1L19. "which leads to the formation of LLJ".

P1L19-20. "thereby reducing the energy demand for the breakdown of this layer".

P1L23. "the passage of a cold front".

…

P2L20-23. "The deep CBL facilitates cloud formation in the late afternoon. This cloud formation not only leads to substantial surface cooling but also causes the momentum in the upper part of the boundary layer to transport downward, resulting in dust emissions".

…

P8L1-2. "This resulted in a high concentration of CNR values being distributed below 1 km, which serves as a material foundation for boundary layer development".

…

P12L1-2. "The persistent high solar radiation resulted in a surface sensible heat flux exceeding 300 W m-2 near the study site (at 16:00 LT), leading to highly efficient atmospheric heating"

…

**Reviewer #2:**

Comments and Suggestions for Authors:

1. The authors did a really thorough and careful job in responding to the comments and revising the summaries. They not only added explanatory materials in the attachment, but also provided a detailed explanation of the algorithm's doubts. The quality of the revised manuscript has improved significantly. I highly recommend accepting this manuscript with just a few minor modifications.

**Response:** Thank you for your recognition of this work. We have made great efforts to improve the quality of the article again.

2. P11 Line29, please correct "more coarse" in "Compared with the CDWL data, the temporal and spatial resolution of the ERA5 reanalysis data is more coarse" as "coarser" or "much coarser".

**Response:** Thank you for your kind reminder. We have rewritten this sentence.

**Change:** P12L8-9. "the ERA5 reanalysis data exhibit coarser temporal and spatial resolution, and the calculation method for the BLH also differs".

3. Authors are advised to standardize the journal name in references by using either the full name or abbreviation as per the specific requirements of ACP journal. e.g., P20 Line17, Line24, Line28, P21 Line2, P23 Line23.

**Response:** Thank you for your kind reminder. We have carefully checked the correctness of the references.

**Change:** P20L14-15. "Browning, K. and Wexler, R.: The determination of kinematic properties of a wind field using Doppler radar, Journal of Applied Meteorology and Climatology, 7, 105-113, 10.1175/1520-0450(1968)007<0105:TDOKPO>2.0.CO;2, 1968.".

P20L19-20. "Che, J. and Zhao, P.: Characteristics of the summer atmospheric boundary layer height over the Tibetan Plateau and influential factors, Atmospheric Chemistry and Physics, 21, 5253-5268, 10.5194/acp-21-5253-2021, 2021".

P20L26-28. "Fiedler, S., Schepanski, K., Heinold, B., Knippertz, P., and Tegen, I.: Climatology of nocturnal low-level jets over North Africa and implications for modeling mineral dust emission, Journal of Geophysical Research: Atmospheres, 118, 6100 - 6121, 10.1002/jgrd.50394, 2013.".

P20L29. "Fujii, T. and Fukuchi, T.: Laser remote sensing, Taylor and Francis Group, 2005.".

P21L4-6. "Guo, J., Miao, Y., Zhang, Y., Liu, H., Li, Z., Zhang, W., He, J., Lou, M., Yan, Y., Bian, L., and Zhai, P.: The climatology of planetary boundary layer height in China derived from radiosonde and reanalysis data, Atmospheric Chemistry and Physics, 16, 13309-13319, 10.5194/acp-16-13309-2016, 2016.".

P21L17-18. "Holtslag, A. and Boville, B.: Local versus nonlocal boundary-layer diffusion in a global climate model, Journal of Climate, 6, 1825-1842, 10.1175/1520-0442(1993)006<1825:LVNBLD>2.0.CO;2, 1993".

P21L19-21. "Hooper, W. P. and Eloranta, E. W.: Lidar Measurements of Wind in the Planetary Boundary Layer the Method, Accuracy and Results from Joint Measurements with Radiosonde and Kytoon, Journal of Climate and Applied Meteorology, 25, 990-1001, 10.1175/1520-0450(1986)025<0990:Lmowit>2.0.Co;2, 1986.".

P21L28-29. "Jones, C.: Recent changes in the South America low-level jet, Npj Climate and Atmospheric Science, 2, 1-8, 10.1038/s41612-019-0077-5, 2019".

P22L9-12. "Ma, Y., Ye, J., Xin, J., Zhang, W., Vilà-Guerau de Arellano, J., Wang, S., Zhao, D., Dai, L., Ma, Yongx., Wu, X., Xia, X., Tang, G., Wang, Y., Shen, P., Lei, Y., and Martin, S. T.: The Stove, Dome, and Umbrella Effects of Atmospheric Aerosol on the Development of the Planetary Boundary Layer in Hazy Regions, Geophysical Research Letters, 47, 1-10, 10.1029/2020GL087373, 2020.".

P22L13-15. "Marsham, J. H., Parker, D. J., Grams, C. M., Grey, W. M. F., and Johnson, B. T. T.: Observations of mesoscale and boundary-layer circulations affecting dust uplift and transport in the Saharan boundary layer, Atmospheric Chemistry and Physics, 8, 8817-8846, 10.5194/ACPD-8-8817-2008, 2008".

P22L28-29. "Ohya, Y., Nakamura, R., and Uchida, T.: Intermittent Bursting of Turbulence in a Stable Boundary Layer with Low-level Jet, Boundary-layer Meteorology, 126, 349-363, 10.1007/S10546-007-9245-Y, 2006.".

P23L27-29. "Washington, R., Todd, M. C., Engelstaedter, S., M'bainayel, S., and Mitchell, F.: Dust and the low-level circulation over the Bodélé Depression, Chad: Observations from BoDEx 2005, Journal of Geophysical Research: Atmospheres, 111, D03201, 10.1029/2005JD006502, 2006.".

P23L30-31. "Wexler, H.: A Boundary Layer Interpretation of the Low-level Jet, Tellus Series A-Dynamic Meteorology And Oceanography, 13, 368-378, 10.1111/J.2153-3490.1961.TB00098.X, 1961.".

P24L3-5. "Wu, G., Wang, T., Wan, R., Liu, X., Li, W., Wang, Z., Zhang, Q., Duan, A., and Liang, X.: The Influence of Mechanical and Thermal Forcing by the Tibetan Plateau on Asian Climate, Journal of Hydrometeorology, 8, 770-789, 10.1175/JHM609.1, 2007.".

P24L6-9. "Wu, K., Wei, T., Yuan, J., Xia, H., Huang, X., Lu, G., Zhang, Y., Liu, F., Zhu, B., and Ding, W.: Thundercloud structures detected and analyzed based on coherent Doppler wind lidar, Atmospheric Measurement Techniques, 16, 5811-5825, 10.5194/amt-16-5811-2023, 2023.".